

# A fire model with distinct crop, pasture, and non-agricultural burning: Use of new data and a model-fitting algorithm for FINALv1

Sam S. Rabin[1,2], Sergey L. Malyshev[3], Brian I. Magi[4], Elena Shevliakova[3], and Stephen W. Pacala[1]

[1]Dept. of Ecology & Evolutionary Biology, Princeton University, Princeton, NJ, USA
[2]Karlsruhe Institute of Technology, Institute of Meteorology and Climate Research / Atmospheric Environmental Research, 82467 Garmisch-Partenkirchen, Germany
[3]GFDL-Princeton University Cooperative Institute for Climate Science, Princeton, NJ, USA
[4]University of North Carolina at Charlotte, Geography and Earth Sciences Department, Charlotte, NC, USA

*Correspondence to:* Sam Rabin (sam.rabin@kit.edu)

**Abstract.** This study describes and evaluates the Fire Including Natural & Agricultural Lands model (FINAL) which, for the first time, explicitly simulates cropland and pasture management fires separately from non-agricultural fires. The non-agricultural fire module uses empirical relationships to simulate burned area in a quasi-mechanistic framework, similar to past fire modeling efforts, but with a novel optimization method that improves the fidelity of simulated fire patterns to new

observational estimates of non-agricultural burning. The agricultural fire components are forced with estimates of cropland and pasture fire seasonality and frequency derived from observational land-cover and satellite fire datasets. FINAL accurately simulates the amount, distribution, and seasonal timing of burned cropland and pasture over 2001–2009 (global totals: $0.434 \times 10^6$ and $2.02 \times 10^6$ km$^2$ yr$^{-1}$ modeled, $0.454 \times 10^6$ and $2.04 \times 10^6$ km$^2$ yr$^{-1}$ observed), but carbon emissions for cropland and pasture fire are overestimated (global totals: $0.297$ PgC yr$^{-1}$ and $0.712$ PgC yr$^{-1}$ modeled, $0.194$ PgC yr$^{-1}$ and $0.538$

PgC yr$^{-1}$ observed). The non-agricultural fire module underestimates global burned area ($1.66 \times 10^6$ km$^2$ yr$^{-1}$ modeled, $2.44 \times 10^6$ km$^2$ yr$^{-1}$ observed) and carbon emissions ($1.33$ PgC yr$^{-1}$ modeled, $1.84$ PgC yr$^{-1}$ observed). The spatial pattern of total burned area and carbon emissions is generally well reproduced across much of sub-Saharan Africa, Brazil, central Asia, and Australia, whereas the boreal zone suffers from underestimates. FINAL represents an important step in the development of global fire models, and offers a strategy for fire models to consider human-driven fire regimes on cultivated lands. At the

regional scale, simulations would benefit from refinements in the parameterizations and improved optimization datasets.

## 1   Introduction

Vegetation fire is an important force for the Earth system at local, regional, and global scales. It can shape ecosystems (Bond and Kelley, 2005; Staver et al., 2011), affect human health (Johnston et al., 2012; Marlier et al., 2012; Hahn et al., 2014), exacerbate or mitigate anthropogenic climate change (Ward et al., 2012; Ciais et al., 2013), and cause direct economic damage

(Doerr and Santín, 2013; Bryant and Westerling, 2014). Fire occurrence can even affect the likelihood of more burning, through positive and negative feedbacks resulting from fire's impact on weather, climate, and vegetation (Laurance and Williamson,





2001; Balch et al., 2008; Zhang et al., 2008). Anthropogenic climate change and increases in atmospheric carbon dioxide concentrations have already increased – or can be expected to increase – the frequency and severity of burning in some parts of the world, while other regions could see decreased burning (Gillett et al., 2004; Westerling et al., 2006; Flannigan et al., 2009;

Krause et al., 2014)

A full accounting of the importance of vegetation fire to the Earth system at present as well as historically and into the future requires the use of dynamic global vegetation models (DGVMs). These simulate processes of vegetation establishment, growth, mortality, disturbance, and competition at large scales using varying levels of mechanism, which allows the regional- and global-level biogeochemical implications of ecosystem dynamics to be fully estimated. When DGVMs are coupled with models

of the soil, atmosphere, and oceans, the resulting Earth system models (ESMs) even simulate how these major components of our planet interact with and feed back upon one another. To understand the complex nature of fire's role in the Earth system, then, realistic models of vegetation burning must be designed and incorporated into DGVMs.

However, fire does not exist solely at the interface of climate and vegetation. Humans play an important role in regulating the fire regimes of many regions around the world (Flannigan et al., 2009; Bowman et al., 2011; Archibald et al., 2013). This

can come about as a result of many processes, one of which is fire's use as a tool to manage agricultural lands. Croplands can be burned to facilitate planting or harvest; for example, sugarcane is typically burned before being harvested, and farmers in many parts of the world burn their crop wastes in the field after harvest (Yevich and Logan, 2003). Pastures and rangelands often see regular burning to reinvigorate the soil and control non-palatable weeds (Uhl and Buschbacher, 1985; Laris, 2002).

The way people burn croplands and pasture in a given region can differ from how the ecosystems there would burn in the

absence of humans, in terms of both frequency and seasonal timing (Le Page et al., 2010; Magi et al., 2012; Rabin et al., 2015). This is significant for modeling efforts because it suggests a decoupling of agricultural fire from the mechanisms governing non-agricultural fire. For example, whereas the fire regime of southern Mali might naturally be dominated by large burns late in the dry season, humans have imposed a regime of small, scattered early burning to avoid such hard-to-control fires (Laris, 2002, 2011).

Unfortunately, previous development of global fire models has mostly glossed over the distinction between agricultural management burning and other burning. Anthropogenic effects on fire most commonly are modeled as dependent solely on population density, not land use (e.g., Venevsky et al., 2002; Arora and Boer, 2005; Pechony and Shindell, 2009; Thonicke et al., 2010; Li et al., 2012; Melton and Arora, 2016; Hantson et al., 2016; Rabin et al., 2016). Moreover, the effect of population density is only to increase or decrease the amount of fire relative to that which would occur naturally – not to affect the intra-

annual timing of fire. There are a few exceptions. The LPJ-LMfire model (Pfeiffer et al., 2013) includes functions to simulate how pre-industrial societies could manage cropland and pasture using fire, but these depend on assumptions that may not apply as well to modern agricultural practices. A fire model developed for the Community Land Model (CLM) by Li et al. (2013) simulates cropland fire, with annual burned area based on socioeconomic data (population density and gross domestic product) and timing based on observations, but pasture is not simulated as a land cover/use type distinct from grassland. The HESFIRE

model (Le Page et al., 2015) accounts for how the amount of human land use (cropland and urban areas) affects burning, but



again pasture is not considered. Neither of these latter two models, moreover, take into account how human activity can affect the *timing* of fire.

To some extent, the neglect of pasture burning in particular – or its convolution with non-agricultural burning – can be attributed to a lack of data. Cropland and a number of other vegetation types can, like fire, be algorithmically mapped using

medium-resolution satellite imagery. Overlaying maps of vegetation type and burned area allows the generation of observational datasets of fire activity on different land covers (e.g., Giglio et al., 2010). However, no such map of global pasture distribution exists – only maps at relatively coarse resolutions describing the fraction of each gridcell that is pasture (e.g., Ramankutty et al., 2008; Klein Goldewijk et al., 2010). When developers of global fire models have designed and parameterized models of non-agricultural burning, they have thus been limited in their choice of observational data with which to

constrain their models. The options have been to either focus on regions with low fractions of cropland and/or pasture (thus potentially biasing their parameterization towards parts of the world inhospitable to agriculture) or to use a dataset "contaminated" with signals from cropland and/or pasture burning. Recently, however, Rabin et al. (2015) used a statistical method to estimate burned area associated with cropland, pasture, and non-agricultural lands at regional scales based on observations of total burned area and estimated land use/cover distributions. This presents an opportunity to create a fire model that not only

explicitly simulates burning practices on cropland and pasture, but also to develop a model of non-agricultural burning based on a purer observational signal.

However, the choice of reference data is only the first step in model development. Model fitting, also referred to as optimization or parameterization, is also critical, and many different methods can be used. Empirical fire models have often been fitted against observations of weather, climate, vegetation state, and anthropogenic factors using regression-type methods (e.g.,

Archibald et al., 2009; Lehsten et al., 2010) or multidimensional search algorithms (Knorr et al., 2014). However, because these methods treat fuel availability as an independent variable, they ignore how fire affects the fuel available for future burning. This fire-biomass feedback can be accounted for by running the fire model interactively with vegetation for parameterization purposes. This process is performed in combination with data from the literature when possible, but it is rather manual and based on trial and error. Ideally, model fitting would combine the best parts of these two approaches to algorithmically search pa-

rameter space for the "best" set of values based on how the model actually performs. Le Page et al. (2015) recently used the Metropolis Markov Chain Monte Carlo method to do just this in fitting the HESFIRE model. This standalone model accounts for fuel availability indirectly, with parameterizations based on precipitation and time since fire. Unfortunately, because of the need for high numbers of iterations, this method cannot be feasibly applied in fire models that are coupled with computationally expensive DGVMs.

Here we describe the development and performance of a DGVM-coupled fire model that uses the new disentangled estimates of burned area associated with cropland and pasture (Rabin et al., 2015) to enable true separation of fire patterns and processes between non-agricultural and agricultural land. A module for non-agricultural fire is fit against the purer, non-agricultural burning data – i.e., observational estimates excluding fire on cropland and pasture – using an algorithm that explores parameter space interactively with the fire and vegetation model. Cropland and pasture fire are explicitly simulated – for the first time, in the case of modern-day pasture fire – by a different module using derived climatologies.





## 2 Fire model

The Fire Including Natural & Agricultural Lands (FINAL) model comprises two different sub-models, simulating separately fire on agricultural and non-agricultural land. Here we describe the model's structure, beginning with the land and vegetation model within which it has been developed, then detailing the separate setups used for simulating non-agricultural and agricultural fire, and finally explaining the simulation of fire's effects on vegetation.

### 2.1 Land and vegetation model

The land model LM3, run by the National Oceanic & Atmospheric Administration Geophysical Fluid Dynamics Laboratory (NOAA-GFDL), is a state-of-the art global dynamic vegetation and land surface model that can be run either offline or interactively with atmosphere and oceans in the GFDL Earth System Model (Shevliakova et al., 2009; Dunne et al., 2013). It simulates five different live plant biomass pools: leaves, heartwood, sapwood, labile carbon, and fine roots. The "stem" biomass pool is

comprised of the heartwood, sapwood, and labile carbon pools. One of five different plant "species," representing biome types with different physiological properties, is assigned to each point based on bioclimatic envelopes and amount of biomass.

One of the most interesting features in LM3 is that it uses sub-gridcell units called tiles, which allow land in different land use types (and in different stages of recovery from land use) to have distinct simulated vegetation and soil. Gridcells can have one each of "natural," cropland, and pasture tiles, along with several "secondary" tiles representing land in different stages of

recovery from wood harvesting or agricultural abandonment. Other, non-vegetated tiles represent glaciers and lakes. Tiles are not spatially arranged, instead existing effectively as a list within each gridcell. Wood harvest and land use transitions occur once per year. At the same time, secondary tiles are merged together if they have similar amounts of heartwood biomass; this prevents the computational burden from becoming unreasonable.

The tiled structure of LM3 could allow it to simulate the heterogeneity of vegetation that fire can create across a landscape,

and cropland and pasture tiles could have fire occur in a completely different way than non-agricultural tiles. The original LM3 fire model did not burn cropland and pasture at all; elsewhere, fire happened once per year based on fuel loading, drought, and historical fire frequency (Shevliakova et al., 2009). The next two sections will describe the structure of the new fire models developed for non-agricultural (natural and secondary; Sect. 2.2) and agricultural (cropland and pasture; Sect. 2.3) tiles.

### 2.2 Burned area: Non-agricultural land

The fire model for non-agricultural lands is based on that developed for the Community Land Model (CLM) by Li et al. (2012, 2013). Total burned area ($BA$) in the natural and secondary fire model is calculated as the product of the number of fires ($N_{fire}$) and burned area per fire ($BA_{pf}$):

$$BA = N_{fire} \times BA_{pf}. \tag{1}$$





### 2.2.1 Number of fires

Lightning and humans both serve as sources of ignitions, some fraction of which actually become fires. Li et al. (2012) modeled

their equation for the density of lightning ignitions after that elaborated by Prentice and Mackerras (1977). At each time step, the number of ignitions from lightning ($I_n$, ignitions $km^{-2}$) is a function of latitude ($\Lambda$, radians) and the density of lightning flashes ($L$, flashes $km^{-2}$):

$$I_n = L \times (5.16 + 2.16cos[3\Lambda])^{-1}. \tag{2}$$

The number of anthropogenic ignitions ($I_a$, ignitions $km^{-2}$) is a function of population density (people $km^{-2}$):

$$I_a = (\beta_{Ia} \times P_D) \times (6.8 \times P_D^{-0.6}) \tag{3}$$

With $\beta_{Ia}$ representing the rate of ignitions per person at each time step and $P_D$ representing population density (people $km^{-2}$), the first part of Equation 3 gives a starting value for density of anthropogenic ignitions per time step. (Henceforth, $\beta$ will denote parameters determined during our optimization routine as described in Sect. 2.6. The final values of these parameters can be found in Table 3.) The second part of Equation 3 is intended to represent the fact that each person can be expected to light

fewer fires as population density increases (Venevsky et al., 2002).

To calculate the number of ignitions actually becoming fires ($N_{fire}$), the total number of ignitions ($A_T[I_n + I_a]$, where $A_T$ is the area of the tile in $km^2$) is multiplied by five functions that vary from zero to one, representing the suppressive effects of relative humidity ($f_{RH}$), soil moisture ($f_\theta$), aboveground biomass ($f_{AGB}$), temperature ($f_T$), and population density ($f_{P_D}$):

$$N_{fire} = A_T(I_n + I_a) \times f_{RH} \times f_\theta \times f_{AGB} \times f_T \times f_{P_D}. \tag{4}$$

Li et al. (2012) calculate the effect of relative humidity on number of fires as

$$f_{RH} = max\left(0, min\left[1, \frac{0.7 - RH}{0.7 - 0.3}\right]\right), \tag{5}$$

where $RH$ (range 0–1) is the relative humidity in the tile. Relative humidity ceases limiting fire (i.e., $f_{RH} = 1$) below $RH = 0.3$, and it suppresses all fire above $RH = 0.7$. However, the artificial limitation of this formulation to the range $[0,1]$ would cause problems during our parameterization, which requires a continuously differentiable function. Instead we used the

Gompertz function:

$$f_{RH} = \exp\left(-\beta_{RH,1} \times \exp[-\beta_{RH,2} \times RH]\right). \tag{6}$$

This function also varies between zero and one, with the parameter $\beta_{RH,1}$ controlling the location of the curve along the $X$ axis and and $\beta_{RH,2}$ determining the steepness of the function as it decreases from one to zero.

Li et al. (2012) formulate the effect of soil moisture on number of fires as

$$f_\theta = \exp\left(\pi \times \left[\frac{\theta}{\theta_e}\right]^2\right), \tag{7}$$





where $\theta$ is relative soil moisture over the top 5 cm and $\theta_e$ is a parameter determining the soil moisture level where approximately 95% of fires are suppressed. This is a continuously differentiable function, but for consistency we used (like $f_{RH}$) a Gompertz function:

$$f_\theta = \exp\left(-\beta_{\theta,1} \times \exp[-\beta_{\theta,2} \times \theta]\right). \tag{8}$$

In addition to flammability as determined by fuel moisture, Li et al. (2012) calculate the effect of above-ground biomass on number of fires as

$$f_{AGB} = max\left(0, min\left[1, \frac{AGB - AGB_{lo}}{AGB_{up} - AGB_{lo}}\right]\right), \tag{9}$$

where $AGB$ ($\mathrm{kgC\,m^{-2}}$) is the sum of aboveground biomass in the heartwood, sapwood, labile carbon, live leaf, and leaf litter pools. (80% of the total biomass carbon in the heartwood and sapwood pools is assumed to be in the aboveground stem, with the remainder in coarse roots.) The parameters ($\mathrm{kgC\,m^{-2}}$) determine the levels of aboveground biomass below which fire is impossible ($AGB_{lo}$) and above which biomass is no longer limiting ($AGB_{up}$). However, as with $f_{RH}$, the fact that this function is not continuously differentiable would create problems for parameterization, so we used a Gompertz function instead:

$$f_{AGB} = exp\left(-\beta_{AGB,1} \times exp[-\beta_{AGB,2} \times AGB]\right). \tag{10}$$

The effect of temperature on number of fires is calculated as

$$f_T = max\left(0, min\left[1, \frac{T - T_{lo}}{T_{up} - T_{lo}}\right]\right), \tag{11}$$

where $T$ ($^\circ$C) is the temperature of the canopy. The $T_*$ parameters ($^\circ$C) serve the same purpose as the parameters in the original formulation of $f_{AGB}$ (Eq. 9); that is, no fire can occur ($f_T = 0$) at or below $T_{lo}$ and temperature does not limit fire ($f_T = 1$) at or above $T_{up}$. After Li et al. (2013), we set $T_{lo}$ to $-10\,^\circ$C and $T_{up}$ to $0\,^\circ$C. Because we did not include this function in the optimization, we did not convert it to a Gompertz function as we did with $f_{RH}$ and $f_{AGB}$.

The suppressive effect associated with increasing population density on all potential fires (as opposed to just anthropogenic ignitions, as accounted for in Eq. 3) is calculated as

$$f_{P_D} = 1 - (0.99 - 0.98 \times exp[-\beta_{PD} \times P_D]), \tag{12}$$

where $P_D$ is human population density (people $\mathrm{km^{-2}}$). $f_{P_D} \to 0.01$ as $P_D \to \infty$, and $f_{P_D} = 0.99$ where $P_D = 0$, after Li et al. (2012). $\beta_{PD}$ determines the shape of the function between these limits.

Li et al. (2013) also included a suppressive effect of per-capita gross domestic product (GDP) on number of fires. This was based on the idea that relatively wealthy parts of the world might have more valuable property to protect and a better capacity for suppression than less developed regions. However, for several reasons, we chose not to include this function. First, although globally gridded maps of GDP exist for the past 25 years or so (van Vuuren et al., 2007), no existing data sets describe the distribution of economic status before 1990. Second, the functions elaborated by Li et al. (2013) are somewhat ad-hoc, not taking into account other variables that might be responsible for the observed relationships. Bistinas et al. (2014), for example,





showed that an apparent relationship between GDP and burned area (Aldersley et al., 2011) can be better explained as an emergent property resulting from the effect of population density. That result does not deal with GDP per capita, of course, but it does indicate the care that must be taken to avoid confounding variables when modeling fire. We thus declined to include GDP effects on burning in our model.

### 2.2.2 Burned area per fire

Burned area per fire is calculated in the CLM fire model based on an approximation of individual fires having elliptical shapes, with the point of ignition being one focus and the fastest spread occurring along the major axis (Fig. 1; van Wagner, 1969). It is made up of three main components: duration, shape, and rate of spread.

Up to a certain point, fires become more elongated with increasing wind speed. That is, higher winds increase the length-to-breadth ratio $LB$ (Fig. 1):

$$LB = 1 + 10 \times (1 - exp[-0.06W]),\tag{13}$$

where $W$ is wind speed ($\mathrm{m\,s^{-1}}$) at 10 meters above ground level. High winds also increase rate of downwind spread relative to the rate of upwind spread, which can also be thought of as increasing the head-to-back ratio $HB$ (Figure 1). $HB$ is related to $LB$ as

$$HB = \frac{LB + \sqrt{LB^2 - 1}}{LB - \sqrt{LB^2 - 1}},\tag{14}$$

Forward rate of spread ($ROS_f$, $\mathrm{m\,s^{-1}}$) – i.e., spread rate downwind from an ignition – is a function of wind speed, fuel moisture, and vegetation type. Vegetation type ("species" *sensu* LM3) determines the maximum possible rate of spread in a tile. We initially defined maximum rate of spread for each species ($ROS_{max,sp}$) based on similar PFT-specific values used by Li et al. (2012 and Corrigendum): 0.4 $\mathrm{m\,s^{-1}}$ for C3 and C4 grass, 0.3 $\mathrm{m\,s^{-1}}$ for tropical and evergreen trees, and 0.22 $\mathrm{m\,s^{-1}}$ for temperate deciduous trees. However, we included maximum rate of spread for tropical tree and C3 and C4 grass in the optimization ($\beta_{ROStt}$ and $\beta_{ROSgr}$, respectively; Sect. 2.6), so 0.4 $\mathrm{m\,s^{-1}}$ and 0.3 $\mathrm{m\,s^{-1}}$ represent their starting values. Their final values can be found in Table 3.

Note that although Li et al. (2012 and Corrigendum) actually used 0.22 $\mathrm{m\,s^{-1}}$ for all forest types other than needleleaf, we increased the initial value of maximum rate of spread in tropical tree tiles closer to that given by Li et al. (2012 and Corrigendum) for shrub PFTs (0.34 $\mathrm{m\,s^{-1}}$). This was done because the rate of spread in tropical savannas is much higher than that in tropical closed forests (especially moist forests), but LM3 has no "shrub" or "savanna" species, with the result that much of the world's tropical savannas are classified as "tropical tree."

The rate of spread realized by any given fire increases with wind speed towards the limit of $ROS_{max,sp}$ according to the function $g(W)$:

$$gW = \frac{2LB}{1 + HB^{-1}} \times g_0,\tag{15}$$





where

$$g0 = \frac{1 + HB_{max}^{-1}}{2LB_{max}}, \tag{16}$$

Here, $LB_{max} = 11$ and $HB_{max} \approx 482$ are the limits of $LB$ and $HB$ as $W \to \infty$ (Equations 13 and 14).

Fires spread more slowly in wet conditions, so fuel moisture is considered in rate of spread. Li et al. (2012) multiplied rate
of spread by $f_{RH}$ (Equation 5) as well as $f_{RH}(\theta)$, the latter being identical to $f_{RH}$ except with soil moisture ($\theta$) replacing relative humidity ($RH$). However, we substituted $f_{RH}(\theta)$ with $f_\theta$ for simplicity and transparency. Thus, the complete equation for forward rate of spread in FINAL is as follows:

$$ROS_f = ROS_{max,sp} \times g(W) \times f_{RH} \times f_\theta. \tag{17}$$

The final component of burned area per fire is the length of time between ignition and extinction. After Li et al. (2012), we
set fire duration ($d$, seconds) to 24 hours (86,400 s).

$$BA_{pf} = \frac{\pi \times (ROS_f \times d)^2}{4 \times 10^6 \times LB} \times \left(1 + HB^{-1}\right)^2. \tag{18}$$

Li et al. (2013) also include functions that reduce burned area per fire based on population density and GDP per capita. We did not include either of these. The issues with using GDP per capita are described in Section 2.2.1 above. Population density might be considered a more trustworthy and meaningful statistic, but as with the GDP functions, the method used by Li et al. (2013) to describe the effect of population density on fire size was somewhat ad-hoc and did not take into account possible confounding factors. Moreover, our model optimization (Sect. 2.6) would have essentially seen the functions relating population density to number of fires and burned area per fire as one large, complicated function. For simplicity and parsimony, then, we did not include an effect of population density on burned area per fire.

Several limits are imposed on $BA_{pf}$. If the burned area calculated at a time step (i.e., $BA_{pf} \times N_{fire}$) is greater than the area of the tile that has not yet burned that day ($A_{t,un}$), $BA_{pf}$ is adjusted for consistency:

$$BA_{pf} = \frac{A_{t,un}}{N_{fire}}. \tag{19}$$

Moreover, we add a limitation to fire size based on landscape fragmentation, based on the idea that fragmentation of the landscape into burnable and unburnable patches tends to prevent fires from reaching their maximum possible size (Archibald et al., 2009; Hantson et al., 2015). Maximum possible fire size as a function of tile size and fraction unburnable area in the gridcell is modeled after the function described by Pfeiffer et al. (2013):

$$BA_{pf,max} = A_t \times \left(1.003 + exp\left[16.607 - 41.503 \times \frac{A_{g,burnable}}{A_g}\right]\right)^{-2.169}. \tag{20}$$

Here, $A_g$ refers to the area of land (including nonvegetated "land" such as glaciers or lakes) in the gridcell, and $A_{g,burnable}$ refers to the area of vegetated land in the gridcell other than cropland. $BA_{pf,max}$ is calculated at the end of each model day – after burning, tile splitting, and land-use transitions have occurred – and applied to the following day.

Burned area is calculated at every fast time step (30 model minutes) and accumulates throughout each day. At the end of each model day, burning occurs (Sect. 2.4).





## 2.3 Burned area: Cropland and pasture

Burned area on cropland and pasture tiles is estimated in a simpler way than that on natural and secondary tiles. At the beginning of each month, some fraction of each cropland and pasture tile burns according to a mean monthly climatology of burned fraction of cropland and pasture. These gridded climatology maps are based on results from the unpacking analysis

(Rabin et al., 2015), which provided monthly estimates of burned area associated with cropland, pasture, and non-agricultural ("other") land. More detail on these input data is provided in Section 3.2. For simplicity, the data from Rabin et al. (2015) may henceforth be referred to as the data from the "unpacking" analysis, or the "unpacked" data.

## 2.4 Fire effects

Carbon in the leaves, stems, and aboveground litter of a burned tile is combusted (i.e., transferred to the smoke pool; Sect. 2.5)

according to species-specific fractional combustion completeness ($CC$) values based on those used by Li et al. (2012). The remaining non-combusted biomass in leaves, stems, and fine roots is subjected to species- and pool-specific fractional mortality ($M$; i.e., transferred to above- or belowground litter), again based on values from Li et al. (2012). Combustion completeness and mortality values used here can be found in Table 1. Note that although the heartwood and sapwood pools are assumed to be 80% aboveground ("stems") and 20% belowground ("coarse roots"), $CC_{stem}$ and $M_{stem}$ are the same for both above- and

belowground pools. This was necessary because LM3 assumes a constant 80%–20% split. However, fire-killed heartwood and sapwood is transferred to aboveground or belowground litter proportionally.

If less than 1 km$^2$ of a tile burns, the tile's biomass is reduced according to $CC \times BF$ and $(1 - CC) \times M \times BF$, where $BF$ is the burned fraction of the tile. This is the method that has been used by every other global fire model previously developed. However, it does not reflect the reality that an actual fire results in a mosaic where only part of the landscape has been burned.

To better represent this process, when $\geq 1$ km$^2$ burns in a given day, FINAL splits the tile into two new tiles – one burned and one unburned. Biomass on the burned tile is reduced by $CC$ and $(1 - CC) \times M$, while the unburned tile is not affected. This "fire tile splitting" occurs on all land cover types except cropland. The 1 km$^2$ threshold was set to reduce computational demand and avoid calculation errors associated with small tiles.

## 2.5 Other changes

The implementation of daily fire and associated tile splitting necessitated many adjustments to parts of the LM3 codebase not dealing with fire directly. Previously, tiles would only be created and/or merged once per year, and secondary vegetation was the only land type allowed to have multiple tiles within a single gridcell. The code for land transitions needed to be reworked to allow daily splitting and merging. We also changed the code to allow all vegetation types, instead of just secondary land, to have multiple tiles. The criteria for merging tiles were also altered to be based on aboveground biomass available for fire

($AGB$ in Equation 9) instead of heartwood. Moreover, we changed the binning structure by which tiles are determined to have similar-enough biomasses to be merged. Previously, bin edges were located at 0.5, 1, 2, 3, 4, 5, 6, 8, 10, and 1000 kg C m$^{-2}$. To better sample ranges of biomass where fuel is limiting, we replaced the first two bin edges with 0.1, 0.3, 0.5, 0.7, 0.9, and

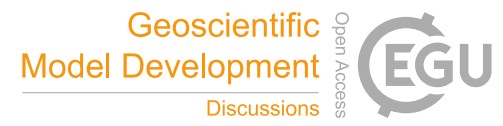



1.1 $\mathrm{kgC\,m^{-2}}$. Finally, various aspects of carbon accounting throughout the model needed to be adjusted for daily tile splitting and merging.

More frequent fire also required other changes. The original LM3 fire module burned once annually at the end of each year, with the burned carbon being emitted gradually over the course of the next year to avoid sudden unrealistic pulses of emissions. With the new fire model operating daily, burned carbon from one day is now emitted over the course of the next day. Previously, grazing of pasture happened once per year, but in order to more reasonably simulate emissions from pasture

fire we made grazing occur daily. We also boosted the fraction of live leaf biomass removed by grazers from $\sim 0.07\%\,\mathrm{day^{-1}}$ to $4\%\,\mathrm{day^{-1}}$ for the main runs (`FINAL_V0` and `FINAL_V1`; Table 2). This resulted in more realistic estimates of aboveground biomass in pasture, and of annual global consumption of biomass by grazers.

Finally, the original LM3 model did not explicitly simulate aboveground dead biomass, which is an important component of the fuel bed in some ecosystems, affecting fire spread and/or emissions. We thus used the version of LM3 with the Carbon,

Organisms, Rhizosphere, and Protection in the Soil Environment model (CORPSE; Sulman et al., 2014), which in addition to simulating the dynamics of soil organic matter also simulates leaf litter and coarse wood litter pools. The default setting for CORPSE is to simulate 15 different belowground soil cohorts (age classes); to improve computational efficiency, we set CORPSE to simulate only one.

## 2.6 Parameter optimization

Simply copying parameters from the model described by Li et al. (2012, 2013) exactly was not possible for a number of reasons. First, here we separately model cropland, pasture, and non-agricultural burning. Li et al. (2013), on the other hand, included special modules for cropland, deforestation, and peat fire – pasture burning being convolved with all other fire. Now that we have extracted from non-agricultural burning the influence of pasture, a significant source of fire activity that often differs from what might be expected under a totally "natural" fire regime, we expect to find different relationships between

fire and its driving variables. Second, CLM is of course a different model than LM3, with its own idiosyncrasies and biases distinct from those of LM3. Although Li et al. (2012, 2013) strove to parameterize their equations based on independent data as much as possible, some functions were entangled with how their model itself worked. Third, as described in Section 2.5, we added some processes and removed others. Fourth, Li et al. (2012, 2013) tested their model against version 3 of the Global Fire Emissions Database (GFED3) burned area dataset (Giglio et al., 2010), whereas we used the GFED3s dataset (Randerson

et al., 2012), which includes an additional estimate of burning from small fires and thus has significantly more burned area than GFED3. Finally, Li et al. (2012, 2013) used different climatic forcing data than we did.

All these differences meant that we needed to reparameterize at least some parts of the non-agricultural fire model. Here we begin by briefly walking through the algorithm used to carry out the optimization, and then describe the parameters that we chose to optimize.



### 2.6.1 The Levenberg-Marquardt algorithm

We used the Levenberg-Marquardt method as the basis of our optimization routine. This algorithm uses the first derivatives of a performance metric with respect to each parameter to iteratively move through parameter space in search of a local minimum of the sum of squared errors. It starts with some initial guess, then evaluates the sum of squared errors $S$ in non-agricultural burned area between the unpacked data and the estimates generated by the model:

$$S = \sum_{m=1}^{M} \sum_{i=1}^{N} (BA_{mod,i,m} - BA_{unp,i,m})^2. \tag{21}$$

(Here, the summation is performed across all $M$ months in the parameterization run period and all $N$ sample gridcells selected for the optimization.) The algorithm then generates a new parameter set guess and the model is rerun. If the new guess decreases the sum of squared errors, it is "accepted," with a new guess then being generated based on it. If not, it is "rejected," and a new guess is generated based on the original guess. Guesses are adjusted by interpolating between steps that would be generated by either the gradient descent method or the Gauss-Newton algorithm, leaning more towards the former when far from a minimum and the latter when near a minimum. More detailed information on the Levenberg-Marquardt algorithm, including its derivation, can be found in Levenberg (1944), Marquardt (1963), and Transtrum and Sethna (2012). Details on our implementation of the algorithm can be found in Appendix A.

The spinup run with which we generated initial conditions for the optimization is described later as `LM3_ORIG` (Sect. 3.1, Table 2). Note that we began the optimization runs in 1991 even though only the 2001–2009 data would be used for comparison to observations; the idea was to allow for the vegetation and fire regime in at least some of the gridcells (especially in regions where frequent fire is the norm) to equilibrate given the fire frequency of each new iteration of the model.

### 2.6.2 Parameters chosen

From the equation for anthropogenic ignitions ($I_a$, Eq. 3), we optimized $\beta_{Ia}$, which can be thought of as controlling a sort of "baseline" value for how many ignitions each person can be expected to provide at each time step. Technically, we optimized $\beta_{Ia,m}$, which is describes the baseline number of ignitions per person per month instead of per timestep (of which there are 48 per day):

$$\beta_{Ia,m} = \beta_{Ia} \times 48 \times \frac{365}{12} \tag{22}$$

All other things being equal, higher values of $\beta_{Ia,m}$ result in more fires.

We also optimized $\beta_{PD}$ from the function describing human suppression of all non-agricultural fires as a function of population density ($f_{PD}$, Eq. 12). All other things being equal, a higher value of this parameter would result in a faster approach of the fraction suppressed towards its upper limit.

Because the LM3 definition of a "species" to describe vegetation type is so broad, we thought it would be especially important to pay attention to several biome-specific maximum rate of spread parameters in FINAL. The "tropical tree" type in



LM3 encompasses a wide range of real-world biomes, from tropical rainforests to semiarid shrublands. The rates of spread for fire in these systems are quite different, and so we included maximum rate of spread in tropical tree regions ($\beta_{ROStt}$) in the optimization. We also included the rate of spread in C3 and C4 grasslands ($\beta_{ROSgr}$), because preliminary testing showed strong overestimates in regions dominated by the C4 grass species especially.

Finally, we optimized parameters from $f_{RH}$ ($\beta_{RH,1}$ and $\beta_{RH,2}$, Eq. 6), $f_\theta$ ($\beta_{\theta,1}$ and $\beta_{\theta,2}$, Eq. 8), and $f_{AGB}$ ($\beta_{AGB,1}$ and

$\beta_{AGB,2}$, Eq. 10). We generated initial guesses for these parameters by fitting Gompertz functions, with the upper asymptote set at 1, to the corresponding functions from Li et al. (2012). Fitting was performed using the MATLAB Curve Fitting Toolbox (MATLAB and Curve Fitting Toolbox Release 2014b, The MathWorks, Inc., Natick, Massachusetts, United States.)

## 3 Experimental setup and analysis

### 3.1 Experimental runs

Spinup of the land to pre-industrial conditions began with a "bare ground" scenario and ran for 300 years, during which climate forcings (Sect. 3.2) from 1948–1977 were repeatedly cycled through. During spinup, atmospheric $CO_2$ concentration was held constant at 286 ppm and land use was turned off. Next, we simulated years 1861–1947, using repeated 1948–1977 climate forcings but historical land use and atmospheric $CO_2$ concentration (Sect. 3.2). Finally, the model was run from 1948–1991 with historical climate forcings, land use, and atmospheric $CO_2$. This run – referred to as LM3_ORIG (Table 2) – provided

initial conditions for other model runs, including the optimization. Note that the daily grazing intensity (Sect. 2.5) was set at its default value of ∼0.07% for LM3_ORIG.

The new model (Sects. 2.2–2.5), with new parameters as described in Section 4.1 and Table 3, was run from 1948–2009 (FINAL_V1; Table 2). This run began with initial conditions as produced for the beginning of 1948 by the original LM3 run described above (LM3_ORIG). An experimental run with the complete new model structure but all settings as initially guessed

in the parameterization (FINAL_V0) was also performed, comparison of which to FINAL_V1 would allow us to explore where the optimization improved or worsened model performance. For both FINAL_V0 and FINAL_V1, daily grazing intensity (Sect. 2.5) was set at 4%.

### 3.2 Input data

The LM3 land and vegetation model is run "offline" in this study, meaning that it is forced by a set of meteorological and

radiation-related variables without any interaction between the land and atmosphere. The variables used here to force LM3 – daily precipitation, surface air pressure, specific humidity, wind vectors, and downward longwave and shortwave radiation – are taken from the observation-based dataset developed by Sheffield et al. (2006). All variables are interpolated to the spatial and temporal resolution of the LM3 fast time step, here set to 30 model minutes. Carbon dioxide ($CO_2$) concentrations are taken from Meinshausen et al. (2011). Historical data on land use transitions and wood harvesting come from the harmonized dataset



created by Hurtt et al. (2011) for use in Earth system models. The mean distributions of cropland, pasture, and non-agricultural land in this study over 2001–2009 are presented in Figure 3.

The FINAL fire model requires additional input data. Cropland and pasture burning (Section 2.3) is forced using climatologies of burned area derived from the unpacking analyses of Rabin et al. (2015), which generated estimates for each of 134 regions around the world based on the GFED3s burned area data (Randerson et al., 2012). The results presented in the main text of that study, with $\widehat{F_k}$ unconstrained, give the net effect of each land-use/cover type on burned area, including any suppressive effects cropland, for example, might have on burned area on non-agricultural land. Here we use the results with $\widehat{F_k}$ constrained to non-negative values, which should provide a more reasonable estimate of how much burning actually occurs on each land

cover type. Note that this method resulted in estimates of total burned area (i.e., burned area summed across all three land cover/use types) slightly greater than the value from GFED3s: $4.93 \, \mathrm{Mha \, yr^{-1}}$ as opposed to $4.68 \, \mathrm{Mha \, yr^{-1}}$. Because the land cover distributions used in the unpacking (Rabin et al., 2015) differ slightly from those used in this study, burned fraction for each gridcell in the unpacked data was adjusted here so that the model output would match the burned area from the unpacking.

For the non-agricultural fire model, we used a gridded monthly climatology of lightning flash rate ($\mathrm{flashes \, km^{-2}}$) based

on data from the Lightning Imaging Sensor (LIS) and Optical Transient Detector (OTD) remote instruments. Specifically, we used the LIS/OTD Low-Resolution Monthly Time Series (LRMTS) described by Cecil et al. (2014). This dataset is provided at a $2.5° \times 2.5°$ resolution, which we interpolated to match the LM3 resolution of $2°$ latitude by $2.5°$ longitude. The version of LRMTS that we used, v2.3, included maps of flash rate for each month in the period 1996–2014. We found the average of each month (January, February, etc.) and used these to build our climatology.

Non-agricultural burning in FINAL also requires input data on population density. We used the historical population density estimates from HYDE 3.1 (Klein Goldewijk et al., 2010), coarsened from their original 5-minute resolution to the LM3 resolution ($2°$ latitude by $2.5°$ longitude). We interpolated population density linearly between each time point in the HYDE dataset.

### 3.3    Evaluation

The new model's performance in terms of recreating observed patterns of burned area and fire carbon emissions is evaluated here by comparison against GFED3s and the unpacked fire data. In addition to global totals of mean annual fire activity, we assess the spatial distribution of fire using maps of mean annual burned fraction and emissions. Unfortunately, due to the short satellite record of fire occurrence, the model must be evaluated against the same time period used for calibration. The model can thus be expected to perform less well outside 2001–2009.

The accuracy of seasonal fire trends is tested by comparing the difference between peak day of burned area simulated by the model with the peak as estimated by the unpacking analysis. This is quantified using mean phase difference, as described by Kelley et al. (2013). Each gridcell's annual pattern of fire can be described as a vector in the complex plane:

$$\boldsymbol{V_i} = (x_{m,i}, \theta_m) , \tag{23}$$





where $x_{m,i}$ is the mean burned area in month $m$ for gridcell $i$, and $\theta_m$ is an arbitrary angle unique to month $m$ and calculated for all gridcells as:

$$\theta_m = 2\pi \frac{(m-1)}{12}. \tag{24}$$

The mean vector $\boldsymbol{L_i}$ for each gridcell has end points that can be described in Cartesian coordinates as the origin and $(L_{x,i}, L_{y,i})$, where:

$$L_{x,i} = \sum_{m=1}^{12} x_{m,i} \cos(\theta_m) \tag{25}$$

and

$$L_{y,i} = \sum_{m=1}^{12} x_{m,i} \sin(\theta_m). \tag{26}$$

The phase ($P_i$), defined where fire occurrence is not distributed evenly across all months, describes the mean timing of peak fire activity:

$$P_i = \arctan\left(\frac{L_{y,i}}{L_{x,i}}\right). \tag{27}$$

The day of the year associated with peak fire activity can be calculated as $\frac{P_i}{2\pi} \times 365$. Mean phase difference $MPD$, which is used here to describe the difference in timing of peak fire between model results and observations, is calculated as

$$MPD = \frac{1}{\pi} \arccos\left(\frac{\sum_{i=1}^{N} cos[P_{i,mod} - P_{i,obs}]}{N}\right), \tag{28}$$

where modeled and observed phases are designated with the subscripts $mod$ and $obs$, respectively. $MPD$ varies from zero to one, with $MPD = 0$ if all modeled peaks correspond exactly to observed peaks and $MPD = 1$ if all modeled peaks differ from observed peaks by the maximum possible amount (6 months).

## 4   Results

### 4.1   Optimized parameters

Figure 4 shows the progression of the parameter guesses, along with the sum of squared errors associated with each parameter set guess through the optimization. The sum of squared errors decreases rapidly for the first few iterations, but diminishing returns become apparent by about the fifth iteration (Fig. 4a). By the eleventh iteration, it did not seem that allowing iterations to continue would result in much improved sums of squared errors, and the optimization was manually halted. The original and

final parameter values can be found in Table 3.

The functions resulting from the new parameter set are visualized, in comparison with how they were in the Li et al. (2012, 2013) model as well as in the initial optimization guess, in Figure 5.


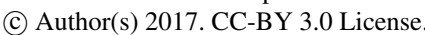


$f_{AGB}$ saw its parameters increase markedly: both $\beta_{AGB,1}$, which translates the function along the $X$ axis, and $\beta_{AGB,2}$, which controls the slope of the increase of $f_{AGB}$ from low to high biomasses (Fig. 4b, c). The net effect relative to the original

guesses was that the amount of fire allowed decreased at biomasses below about $0.3 \mathrm{\,kg\,C\,m^{-2}}$ and increased between about $0.3$ to $1.5 \mathrm{\,kg\,C\,m^{-2}}$ (Fig. 5g).

The parameter controlling anthropogenic ignitions, $\beta_{Ia,m}$, decreased through the sixth guess, then increased to a level higher than initially guessed, before declining again to a low level by the end of the optimization (Fig. 4d). The density of anthropogenic ignitions $I_a$ is thus decreased at all positive levels of population density (Fig. 5a). Moreover, the parameter $\beta_{PD}$ – which controls anthropogenic suppression of burning $f_{P_D}$ – increased (Fig. 4e), meaning that a larger fraction of ignitions (both lightning and anthropogenic) are suppressed wherever population density is greater than zero, though most noticeably between densities of $10$–$100 \mathrm{\,people\,km^{-2}}$ (Fig. 5b). The net effect is to reduce unsuppressed anthropogenic ignitions (i.e.,

$I_a \times f_{P_D}$) relative to the initial guess, with the peak's location being mostly unchanged but its severity being modulated (Fig. 5e).

Four parameters relating to the effect of moisture on fire activity were optimized: $\beta_{RH,1}$ and $\beta_{RH,2}$, which control the effect of relative humidity $f_{RH}$, and $\beta_{\theta,1}$ and $\beta_{\theta,2}$, which control the effect of soil moisture $f_\theta$. Altogether – i.e., taking into account moisture effects on both ignition success probability and rate of spread – burned area in FINAL is proportional to $(f_{RH} \times f_\theta)^3$.

Because $f_{RH}$ and $f_\theta$ always appear together in the model equations, and because relative humidity and soil moisture might be expected to be strongly correlated, one might have expected the optimization to result in similar functions. However, the final shapes of $f_{RH}$ and $f_\theta$ are quite different (Fig. 5c, d). $\beta_{\theta,1}$ increased and $\beta_{\theta,2}$ decreased (Fig. 4h, i), resulting in a stronger suppressive effect of soil moisture: Whereas the original function suppresses nearly all fire beginning at around $\theta = 0.65$, the new function reaches this point around $\theta = 0.35$ (Fig. 5d). Even in extremely dry soils where $\theta = 0$, $f_\theta = 0.7$ – meaning that

around 30% of ignitions would be prevented from becoming spreading fires, and rate of spread would be reduced by 51%. $f_{RH}$, on the other hand, was effectively neutered: While $\beta_{RH,1}$ and $\beta_{RH,2}$ both increased (Fig. 4), $\beta_{RH,2}$ increased so drastically that $f_{RH} \approx 1$ for all values of relative humidity (Fig. 5c). Figure 5f shows that the total effects of these shifts in the moisture functions are most extreme at low values of soil moisture, with low levels of relative humidity burning less and high levels of relative humidity burning more (all other things being equal). However, LM3 never produced the latter condition (Fig. 6d),

and so low-humidity cells seem to have driven this trend. The fact that the optimization took place at a monthly scale may also have contributed to the algorithm's lessening of relative humidity's role (Sect. 5.3).

Maximum rate of spread decreased more than 25% for grassland (Fig. 4k), a result which likely has to do with the model overestimating fire in these low-biomass systems. This parameter decreased sharply for most of the optimization, but as $f_{AGB}$ appropriately began to take on more of the responsibility for regulating fire there, grassland maximum rate of spread began to

increase back towards its initial guess. Maximum spread rate increased by over 300% for the "tropical tree" vegetation type (Fig. 4j), due to a tendency towards underestimation of burned area in that biome.

Comparing the results of FINAL_V0 with FINAL_V1, we can see that much of the improvement came in regions where the initial parameter set severely overestimated burned area (Fig. 7a–d). Performance worsened in other gridcells. A map of root mean squared error (Fig. 7e), which shows performance improvement as would be "seen" by the optimization algorithm



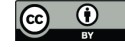

for included gridcells, highlights a few cells in and around the tropical rainforests of Africa and South America as areas where the performance metric increased markedly (indicating worsened performance) between the initial and final guesses. Semi-arid regions tended to show improved performance with the new parameter set, with Figure 7e highlighting northwest Mexico, northern Argentina, Botswana, the periphery of the Sahara, and to a lesser extent the Middle East and Australia. Overall, global RMSE (i.e., the sum of all gridcells' RMSE) decreased from $\sim 7.54 \times 10^5$ ha to $\sim 5.57 \times 10^5$ ha ($\sim$26.1% improvement).

## 4.2  Model performance

Figure 8 compares, over 2001–2009, maps of mean annual burned fraction (i.e., fraction of land area) from run `FINAL_V1` with those from GFED3s (Randerson et al., 2012) and the unpacking analysis. Figure 9a shows the difference in mean annual burned fraction between the model and the unpacked observations, against which the non-agricultural model was parameterized.

Considering all land cover types together, the new fire model recreated the general pattern of annual fire activity well compared with both GFED3s (Randerson et al., 2012) and the unpacked data (Figs. 8a,b,f; 9a). The largest modeled overestimates relative to the unpacked data occurred in the grasslands and shrublands of western South America, the western Caatinga of northeast Brazil, and at various points throughout the African savannas (Fig. 9a). Most of the severe model underestimation relative to the unpacked data occurred in the African tropical savannas, as well as (to a lesser extent) the tropical savannas of northern

Australia (Fig. 9a).

The modeled burned fractions of cropland and pasture match the unpacked numbers almost exactly (Figs. 9c,d), which is not surprising considering that the unpacked data were used to force the model on cropland and pasture tiles. There are some notable discrepancies, however. Specifically, there is too much cropland fire in one European gridcell and too little in several gridcells in northern Australia (Fig. 9c). Pasture fire did not experience such severe error in burned fraction anywhere (Fig. 9d).

The strong correspondence of modeled cropland and pasture fire with the unpacked observations (as expected since the latter were directly used to drive the former) suggests that the majority of the error seen in total burning must be associated with fire on non-agricultural lands. Indeed, although the non-agricultural fire model generally captured the worldwide distribution of fire – with tropical savannas, grasslands, and shrublands generally dominating burned area – the fit is by no means perfect (Fig. 9b). There are a number of regions where the model simulates little to no non-agricultural burning but the unpacked

data show significant amounts of fire (Figs. 8b,f) . This phenomenon is especially noticeable in the eastern African savannas, the shrublands of western Australia, and throughout the tropical and temperate grasslands, savannas, and shrublands of South America.

Worldwide, the non-agricultural fire model underestimated burned area, with $1.66 \times 10^6$ km$^2$ yr$^{-1}$ simulated as having burned – an underestimate of 32% relative to the unpacked estimate. Unsurprisingly given the spatial results presented above,

global averages for cropland and pasture were much better – $0.434 \times 10^6$ km$^2$ yr$^{-1}$ (4% underestimate) and $2.02 \times 10^6$ km$^2$ yr$^{-1}$ (1% underestimate), respectively. Mean annual global burned area across all land covers over 2001–2009 was modeled as $4.11 \times 10^6$ km$^2$ yr$^{-1}$, an underestimate of 12% relative to GFED3s and an underestimate of 17% relative to the unpacked total. The time series of annual burned area over 2001–2009 for each land cover from the model (i.e., `FINAL_V1`) are compared with the GFED3s and unpacked estimates in Figure 10a.



Just as the model tended to underestimate total global burned area, it also underestimated carbon emissions from fire (Table 4). The 2.34 PgC yr$^{-1}$ simulated by the model represents an underestimate of 6% relative to GFED3s and of 9% relative to the unpacking data. This is again principally due to non-agricultural fire, for which the model simulated 1.33 PgC yr$^{-1}$ as opposed to the unpacked estimate of 1.84 PgC yr$^{-1}$ – an underestimate of 28%. Agricultural fire emissions were actually overestimated, with 0.297 PgC yr$^{-1}$ for cropland and 0.712 PgC yr$^{-1}$ for pasture – overestimates of 53% and 32% compared to the unpacked values of 0.194 PgC yr$^{-1}$ and 0.538 PgC yr$^{-1}$, respectively.

5    The spatial distribution of errors in total fire carbon emissions (Fig. 9e) generally reflects the distribution of errors in simulated burned area (Fig. 9a). As with burned area, there are sizable regions where the model simulates little to no non-agricultural fire carbon emissions but the unpacked data show otherwise (Figs. 11e,i). Cropland fire emissions, as with burned area, are underestimated in northern Australia; there are also two regions in central Africa where cropland fire emissions are overestimated despite essentially correct annual burned fraction (Figs. 11c,g). The areas of slightly underestimated pasture burned fraction are not apparent in the map of pasture fire emissions error; large overestimates of emissions from pastures in the tropical savanna biome are instead the most apparent aberrations (Figs. 11d,h).

The non-agricultural fire model performed well in terms of simulating the within-year timing of burned area (Figs. 12e,i). This was reflected in the results for combined burning across all land cover types, which corresponded well with both GFED3s and unpacked burned area (Figs. 12a–b,f); the timing of peak model-estimated fire was 35 days later than observed for all fire combined as compared with total unpacked fire (mean phase difference $MPD = 0.19$), and 53 days later than observed for non-agricultural fire specifically ($MPD = 0.29$).

## 5    Discussion

### 5.1    Model performance in context: Burned area

In terms of spatial distribution, the model tends to over-cluster non-agricultural burned area relative to the unpacked estimate. That is, it tends (especially in savanna regions) to simulate a highly spatially heterogeneous distribution of non-agricultural burned area, with some areas burning very little and others burning far too much (Fig. 8). It is important to consider, however, that although the unpacking method generates accurate estimates of total burned area at the level of each analysis region, the burning tends to be too evenly distributed within each region (Rabin et al., 2015). This results in an overly smooth map, as can be seen by comparing maps A and B in Figure 8. Non-agricultural burning in the real world might thus exhibit more spatial clustering than is apparent in Figure 8e. To get a sense of the spatial clustering of real-world non-agricultural fire, we have constructed a map of mean annual "GFED3s non-agricultural" burned fraction by subtracting unpacked cropland and pasture burned fraction from mean annual GFED3s total burned fraction. (The exact numbers from this map are not very meaningful, since it is possible to have values less than zero in gridcells where unpacking estimated more cropland and pasture burning than all burning observed by GFED3s; the purpose of this exercise is only to examine spatial heterogeneity.) A map of the coefficient of variation in 6×6 gridcell (12° latitude × 15° longitude) kernels across this map is compared with similar maps for mean annual modeled and unpacked non-agricultural fire in Figure 13. As expected, the coefficient of variation is much higher



in the GFED3s data than the unpacked data, indicating stronger spatial clustering of non-agricultural fire in the real world. The fact that the model simulates more heterogeneity than the unpacked estimate, then, indicates that the model is capturing heterogeneity in fire drivers that are important to actual fire patterns. This is not to say, of course, that the heterogeneous patterns simulated by the model exactly match the observations – in some places they do not, as is apparent in Figure 8.

Although savanna regions may have shown the largest absolute difference in modeled vs. unpacked fire activity, smaller differences can be just as important in other areas. For example, the GFED3s and unpacked data show a mean annual burned fraction of 1–5% for the boreal forests of central Alaska and northwestern Canada (Figs. 8a–b,e), which would correspond to a mean fire return interval of 20–100 years. While this is a low rate of burning relative to, e.g., tropical savannas, it still represents an important process for the structure and function of that ecosystem. The non-agricultural fire model captures almost no boreal forest fire whatsoever (Fig. 8i), which should hamper the ability of LM3 to accurately simulate vegetation there. One possible contribution to this deficit is the importance of multi-day fires in the boreal region. We followed Li et al. (2012) in assuming that all fires last 24 hours, but this assumption is not well-supported by the literature. Korovin (1996) found that almost 60% of forest fires in Russia over 1947–1992 lasted longer than one day, and that fires lasting longer than 10 days accounted for nearly 70% of the burned forest area. Stocks et al. (2003) found a similar importance of very large (and thus presumably long-lasting) fires in Canada, with individual burns of more than 20,000 ha comprising over 65% of mean annual burned area over 1959–1997. Ideally, FINAL would replicate this pattern by explicitly modeling the duration of individual fires based on evolving weather conditions. Several global fire models have introduced such a component, but with mixed results. The LPJ-LMfire model developed by Pfeiffer et al. (2013), which allows fires to burn for about four hours per day until they experience significant precipitation, actually tends to *overestimate* boreal forest fire. The HESFIRE model (Le Page et al., 2015) also allows fires to burn indefinitely, calculating twice per day an extinction probability based on fuel load, attempted suppression intensity, landscape fragmentation, and weather conditions. However, like FINAL, HESFIRE simulates too little fire in the boreal region (Le Page et al., 2015).

## 5.2 Model performance in context: Emissions

The tendency of `FINAL_V1` to underestimate total global 2001–2009 burned area is reflected in an underestimate of the associated carbon emissions – by 6% and 9%, respectively, relative to GFED3s (Table 4). GFED3s and the unpacking data show respective average emissions densities of 0.53 and 0.52 $\mathrm{kgC\,m^{-2}}$ of burning for all fire combined, whereas `FINAL_V1` gives 0.57 $\mathrm{kgC\,m^{-2}}$ (based on Table 4). The largest discrepancy in fire carbon emissions density between the modeled and unpacked estimates is on cropland, where `FINAL_V1` simulates 0.68 $\mathrm{kgC\,m^{-2}}$ but the unpacking analysis gives only 0.43 $\mathrm{kgC\,m^{-2}}$ (58% overestimate; Table 4). Emissions densities on pasture and non-agricultural land are also overestimated, respectively by 35% and 6.7%.

Given how extensive pasture burning is at a global scale, it is especially important to understand why the C emissions density of pasture fire was so significantly overestimated. Emissions from pasture fires, as with all fires, are the product of three quantities: burned area, aboveground biomass, and combustion completeness. Because the model simulates burned pasture area so accurately (Table 4), either or both of the latter two could have contributed to the overestimation of pasture fire emissions.




It is important to keep in mind that the records of fire emissions in the GFED product are not purely observation-based. GFED emissions estimates are generated by forcing a version of the Carnegie-Ames-Stanford-Approach (CASA) model with GFED burned area, using vegetation type and soil moisture to determine combustion completeness (van der Werf et al., 2006, 2010). Biases may exist in that model that result in incorrect estimates of aboveground biomass and/or combustion completeness. Apparent discrepancies between GFED3s and FINAL-simulated fire emissions thus may not represent true errors by FINAL relative to reality. Here, we compare aspects of LM3 and FINAL with regard to pasture biomass; this allows us to not only test

whether FINAL appears to be overestimating actual pasture fire emissions, but if so, to also diagnose possible causes.

On average over 2001–2009, FINAL_V1 simulated 3.4 $\mathrm{kgC\,m^{-2}}$ of aboveground biomass on pastures, including both live vegetation and dead material. This was broken down into live leaves (0.22 $\mathrm{kgC\,m^{-2}}$), live stems (0.94 $\mathrm{kgC\,m^{-2}}$), leaf litter (0.45 $\mathrm{kgC\,m^{-2}}$), and dead woody material (1.8 $\mathrm{kgC\,m^{-2}}$); these pools are mapped for the world's major pasture regions in Figure 14. In their work in the Waikato region of New Zealand – a moist, temperate ecosystem dominated by C3 grasses –

Hanna et al. (1999) defined active pastures as containing no more than 0.2 $\mathrm{kgC\,m^{-2}}$ of live leaves or 0.15 $\mathrm{kgC\,m^{-2}}$ of dead material. FINAL_V1 simulated less than 0.1 $\mathrm{kgC\,m^{-2}}$ of live leaf tissue in New Zealand, and indeed the world's temperate pastures seem to satisfy the $\leq 0.2\ \mathrm{kgC\,m^{-2}}$ criterion (Fig. 14a). The tropics generally see much higher modeled pasture leaf biomass; in all cases, leaf biomass does not much exceed 0.25 $\mathrm{kgC\,m^{-2}}$ (Fig. 14a). Uhl and Kauffman (1990) describe a pasture in eastern Amazonia with 0.6 $\mathrm{kgC\,m^{-2}}$ of nonwoody material; this is close to the simulated value of combined live

and dead leaf C (Fig. 14a,c) in the regions listed above. Kauffman and Cummings (1998), looking at three other pastures in Amazonia, found a range of 0.8–1.5 $\mathrm{kgC\,m^{-2}}$ of fine fuels, which included both live and dead leaf material as well as fine woody debris. Again, this corresponds well with our results (Fig. 14a,c), although we do not simulate fine woody debris. Kauffman and Cummings (1998) also found 1.3–5.2 $\mathrm{kgC\,m^{-2}}$ of large downed trunks remaining from the initial clearance of forest for pasture; the simulation produces levels of woody litter in that range for pastures in the Atlantic Forest region of

Brazil and in southern China (Fig. 14d).

LM3 does seem to have overestimated pasture biomass in tropical savanna regions, however. Savadogo et al. (2007) found a mean of 0.045 $\mathrm{kgC\,m^{-2}}$ in the tree and bush savanna of Burkina Faso, where LM3 using FINAL_V1 simulates live biomass pools (leaf + stem) of up to about 0.5 $\mathrm{kgC\,m^{-2}}$ (Fig. 14a, b). Savadogo et al. (2007) also found a mean of 0.07 $\mathrm{kgC\,m^{-2}}$ of dead material there, whereas our model simulated values of around 0.2–0.3 $\mathrm{kgC\,m^{-2}}$ (Fig. 14c, d). Ottmar et al. (2001) found

that land in the Cerrado with a significant herbaceous layer (*campo limpo*, *campo sujo*, and *cerrado ralo*) generally tended to have less than 1 $\mathrm{kgC\,m^{-2}}$ of aboveground live and dead biomass; our model simulated about 1–1.5 $\mathrm{kgC\,m^{-2}}$ (Fig. 14e). It is not clear whether the sites examined by Ottmar et al. (2001) were actively grazed; if not, pastures there would be expected to have even less biomass, in which case LM3's overestimate would be more pronounced.

A widespread overestimation of biomass in tropical savannas would at least partially explain the tendency toward overesti-

mated pasture fire carbon emissions there (Fig. 9d, h). Because most of the world's pasture fire occurs in this biome (Fig. 8), it would also explain the 32% overestimate of mean annual global pasture fire carbon emissions (Table 4). Excess simulated plant matter in tropical savannas could result from any or all of several factors. It is possible, for example, that grazing intensity is unrealistically low.





LM3 does not appear to have simulated too little grazing at a global level. With the rate of grazing set to 4% of leaf biomass each day, the `FINAL_V1` run simulated the consumption by livestock of $1.54\,\mathrm{PgC\,yr^{-1}}$ globally over 2001–2009. This compares favorably with previously-published estimates of carbon flows to livestock. Wirsenius (2000) estimated that domesticated grazers consumed $1.33\,\mathrm{PgC}$ in 1990, not counting draft animals. Krausmann et al. (2008), working on the year 2000, estimated that livestock (including draft animals) consumed $1.9\,\mathrm{PgC}$. Haberl et al. (2007) estimated that the average grazing pressure on pasture for the year 2000 was $41\,\mathrm{gC\,m^{-2}}$, which again compares favorably with the simulated value from `FINAL_V1` of $45\,\mathrm{gC\,m^{-2}\,yr^{-1}}$ over 2001–2009.

Although the global amount of grazed vegetation seems to have been simulated well (as discussed above), much variation likely exists among regions in how intensely land is grazed. This is not captured by the assumption in our model of a 4% daily grazing rate. Combustion completeness values being too low would also lead to too-high estimates of aboveground biomass, but the possible effect of this on estimated emissions is unclear. Increasing combustion completeness would increase fire emissions in the short term, but as any individual pasture tile grew older and approached equilibrium biomass, fire emissions might be no different. That is, decreased biomass with increased combustion completeness might not change emissions density.

Lastly, the fact that FINAL does not explicitly simulate fire associated with land clearance likely contributes to its over-estimation of cropland and pasture fire emissions density. In the version of LM3 used here, biomass killed during land use transitions can be either harvested or wasted. Harvested wood biomass goes to one of three long-lived virtual emissions pools, while wasted biomass is transferred to litter. But in reality, wood remaining after harvest (also known as slash) is often burned, especially in the high-biomass moist tropical forest biome. The emissions involved are significant: Tropical deforestation burns were estimated by van der Werf et al. (2010) to contribute up to 15% of global annual fire $CO_2$-C emissions on average. Instead of breaking this out into a separate flux, LM3 and FINAL are conflating land clearance fire emissions with the emissions from subsequent burning of the cleared land for agricultural management. This is unfortunately not a mere accounting quirk; the use of one or two burns to get rid of most of the remaining slash wood means that fire emissions spike soon after land clearance, whereas LM3 and FINAL simulate a gradual decrease over time. However, the frontier regions of moist tropical forests do not exhibit as much error in cropland and pasture fire carbon emissions as is seen in tropical savannas (Fig. 9e,f), and so the relative importance of this model behavior to simulated carbon fluxes at a global scale appears to be limited.

### 5.3 What do optimization results suggest?

The optimization effectively excluded relative humidity from exerting any effect on fire activity, shifting all of the control of flammability to soil moisture (Fig. 5c, d). This suggests that, at the coarse spatiotemporal scale considered, the moisture of the upper soil may be a much better proxy for fuel moisture than relative humidity. This could represent a real phenomenon: Live fuels such as the herbaceous layer in grasslands and savannas have access to soil water that, even in the upper soil, likely fluctuates less over short time scales than relative humidity. Where live vegetation and/or slow-drying coarse woody debris are a major part of the fuel bed, then, soil moisture might be a better proxy of fuel moisture. But in the real world, humidity is often a good predictor of flammability, and operational fire danger indices usually include it (e.g., Noble et al., 1980). The exclusion of relative humidity as a predictor may only have emerged here as an artifact of our optimization structure. Humidity does exert



some control on fuel moisture at fast time scales, but our algorithm evaluated model performance on a month-by-month basis. Soil moisture may do a better job of tracking seasonal trends in flammability that are relevant at that time scale. If instead we had performed a comparison at daily scale, relative humidity might have proven important.

The fact that the soil moisture suppressive effect does not abate even for the driest soils – that is, $f_\theta(\theta = 0) \approx 0.7$ instead of 1 (Fig. 5d) – is another intriguing result. Because $f_\theta(\theta = 0) = exp(-\beta_{\theta,1})$ (Eq. 8), it would have been reasonable to constrain $\beta_{\theta,1}$ during the optimization to prevent $f_\theta(\theta = 0)$ from being below 0.999 or some other value close to unity. Such a strategy

would arguably even make physical sense – soil moisture can hardly limit fire if there is no moisture in the soil. This would presumably have the effect of increasing burned area at low soil moisture, but that might not be the case. It's possible that very few gridcells ever actually experienced such low soil moisture, and/or such cells were limited by other factors – chronically low soil moisture (or average conditions in regions that ever experienced such an extreme) would result in low aboveground biomass, for example. If true, this could mean that the result of $f_\theta(\theta = 0) \approx 0.7$ may essentially have been spurious, since the

algorithm would not have been very sensitive to $f_\theta$ at such low values of soil moisture. On the other hand, this might be a real effect, in which case there may be a more structural issue with the fire model. A simple scaling factor – some extra constant that reduces ignition density, for instance – could be a useful addition in that case, but would have the function of decreasing fire in all gridcells.

At the other end of the soil moisture function, moistures above ∼0.35 prevent almost all fire from occurring, whereas the

initial guess didn't restrict so severely until about $\theta = 0.65$ (Fig. 5d). Le Page et al. (2015), in the manual phase of their model development, decided that soil moisture would prevent all burning above $\theta = 0.35$ as well. Although soil moisture in that model only affected rate of spread and not also ignition success rate as it does in our model, and although they also allowed relative humidity to affect rate of spread in a manner similar to Li et al. (2012), the fact that our optimization's result corresponded so closely with their parameter choice is intriguing. However, inspection of model output (not shown) indicates that the soil

moisture function may have contributed to the underestimation of fire in the boreal zone and in the savannas of Zambia and the southern Democratic Republic of the Congo: No month in 2001–2009 had a mean soil moisture <0.35 across much of those regions.

Optimization resulted in fewer anthropogenic ignitions and stronger anthropogenic suppression for any given value of population density (Fig. 5a–b, e). This suggests that, by grouping together non-agricultural fires with pasture fires, previous mod-

eling efforts may have overestimated the contribution of humans to burning on non-agricultural land. That is, by extracting a "pure" non-agricultural fire signal, our study shows that pasture burning practices may have been responsible for much of what was once characterized as general anthropogenic fire, and that humans enhance fire on non-agricultural lands less than once believed. In terms of the general shape of net anthropogenic influence on non-agricultural fires – including the location and width of the peak – our results do not differ substantially from the function described by Pechony and Shindell (2009) or that

used by Li et al. (2012; Fig. 5e). Knorr et al. (2014), on the other hand, used the Levenberg-Marquardt algorithm to fit a simple empirical fire model in a non-interactive fashion and found that the peak was actually located closer to a population density of $0.1 \, \text{people} \, \text{km}^{-2}$ than to the value of $\sim 10 \, \text{people} \, \text{km}^{-2}$ that we found here.





When considering the results of this optimization, it is important to keep in mind that even if the Li et al. (2012, 2013) model had been used in LM3 without modification, performance would have differed from the original CLM version. Structural differences between CLM and LM3 result in different vegetation dynamics and micrometeorology relevant for fire. We also calibrated our model based on different input data and observations than those used by Li et al. (2012, 2013). These and other differences create uncertainty about exactly why any given function's parameters shifted as they did during our optimization. The Fire Model Intercomparison Project (FireMIP; Rabin et al., 2016) could be informative in this regard.

## 5.4 Levenberg-Marquardt optimization: Lessons learned

One of the limitations of the Levenberg-Marquardt algorithm is that it can only "move downhill." At every iteration, it searches for new parameters in the direction of lower sum of squared errors from the current point in parameter space, even though the set of parameters with the lowest possible sum of squared errors may be in a totally different direction. As an analogy, imagine a person given the task of finding the lowest point in a city. Using a "downhill-only" algorithm, this person would literally walk downhill from their starting point and stop when they reach a point – the local minimum – where continued travel in any direction would be uphill. The person might more thoroughly search the city for its lowest point by occasionally turning uphill and/or randomly taking a bus once in a while to a totally different part of the city – analogous to the behavior of the Metropolis-Hastings or simulated annealing algorithms.Levenberg-Marquardt being a downhill-only algorithm is not a fatal flaw, especially when the initial parameter set guess is well-informed based on the literature. It may well represent an improvement in methodology over the manual trial-and-error approach. But it is important to remember that Levenberg-Marquardt should not be expected to produce the universally best possible parameter set.

Another, potentially more serious limitation of the Levenberg-Marquardt algorithm is its use of the sum of squared errors (SSE) as a metric to gauge model performance. While the setup used here does account for accuracy of burned area simulations in both space and time, SSE tends to result in a bias towards improving performance in gridcells where the model simulates burned areas much higher or much lower than observations. This tendency to reduce absolute error would be fine if the goal of optimization were to produce a model that accurately simulates burned area for its own sake, but *relative* error can be more reflective of how well the model simulates the state of the vegetation. For example, assume two hypothetical 1,000-$km^2$ gridcells: one dominated by tropical grassland where observations show 100% annual burning but the model simulates 25%, and one dominated by boreal forest where observations show 1% annual burning but the model simulates 0.25%. In both cases, the model is producing 75% less fire than what actually happens – a difference that could be extremely important to the simulated structure and function of both ecosystems. However, because the absolute error in the grassland gridcell ($-750$ $km^2\,yr^{-1}$) is so much greater than that in the boreal forest gridcell ($-7.5$ $km^2\,yr^{-1}$), the former will, all other things being equal, have a much greater influence on the direction and magnitude of the step towards the next parameter set guess. Our use of an equirectangular grid – with cells of constant size in terms of latitude and longitude but not physical area – means that cells from high latitudes are much smaller than cells from the tropics, which exacerbates this issue. Because the observations show that tropical savannas burn far more than any other biome, the absolute errors are highest there (Fig. 9). These regions thus likely drive most of the optimization, which could have led to the neglect of performance in, for example, the boreal



region. An optimization algorithm that took relative error into account might thus improve performance in low-fire regions, while worsening it where fire is frequent.

Simply substituting an alternative measurement for SSE in a Levenberg-Marquardt context would be less than ideal for addressing this problem. In addition to being the performance metric – i.e., the statistic by which the algorithm determines whether a parameter set has resulted in improved model performance – SSE is an inherent part of the mathematics in the Levenberg-Marquardt algorithm generating the direction and size of the step from the most recently accepted guess to the next accepted guess (Levenberg, 1944; Marquardt, 1963; Transtrum and Sethna, 2012). Using a different performance metric would

still result in guesses designed to minimize SSE. This would at best reduce the efficiency of the algorithm, and at worst result in searches orthogonal to the direction of improved performance. To most effectively avoid the problems inherent with SSE, a completely different algorithm – preferably one that can use any arbitrary performance metric – would be needed. The Markov Chain Monte Carlo method (MCMC) is one such option, which has the additional benefit as discussed above of being a global search algorithm. It has been widely used in the Earth sciences, including by Le Page et al. (2015) to fit a global fire model.

Those authors used as their performance metric a combination of (a) accuracy of classification of gridcells into burned fraction bins and (b) level of correspondence between model-simulated and observed interannual variability. However, being a global search, MCMC requires many iterations to converge on an optimal solution – Le Page et al. (2015) reported iteration counts of hundreds to over a thousand. The deeply model-interactive setup used here – where the complete model of soil, vegetation, and fire was forced with climatic data for 19 model years – took around two hours per iteration with all gridcells being run in

parallel, which made MCMC and similar many-iteration algorithms computationally infeasible.

The choice of gridcells and initial conditions is also extremely important to any automated model fitting algorithm. The strong effects we saw in preliminary optimization runs of including a few extra gridcells from badly-modeled regions make this quite clear. The process through which we settled on our set of 241 gridcells was admittedly haphazard, and a more structured and informed approach would likely make the results more robust. Similarly, we did not experiment with different

initial parameter set guesses, but doing so is a good way to test model robustness (Knorr et al., 2014; Le Page et al., 2015).

## 6   Conclusions: Regional variations in pattern and practice

FINALv1 represents the first attempt in an Earth system model to separate present-day cropland, pasture, and non-agricultural burning. The importance of this can be seen, for example, in differences between pasture and non-agricultural land in the timing of peak burned area – especially in central Asia (Fig. 12). These land use/cover types also differ in fire frequency, as exhibited

for example in northern Australia (Fig. 8). Overall, the combined fire model tends to perform well over much of sub-Saharan Africa, Brazil, central Asia, and Australia. However, non-agricultural burning specifically is not well-represented in several important regions; these include eastern sub-Saharan Africa, South American savannas and grasslands, interior Australia, South and Southeast Asia, and the boreal zone (Fig. 8). A strong limitation of fire by soil moisture may have much to do with performance in those parts of the world (Section 5.3). The apparent deficiencies of the non-agricultural fire module – the first to



be tested against globally gridded estimates of non-agricultural burning – may reflect the need to more fundamentally rethink how non-agricultural fire is represented in global models.

The use of climatologies for cropland and pasture burned area is a significant limitation on FINALv1. It allows very little interannual variability (Figure 10) – only what results from changing agricultural area. Perhaps more importantly, however, the use of a climatology based on just nine years of observations makes it difficult to justify the use of the model very far into the past or future. Economic development can result in changes in technology, types of crops, and legislative priorities (banning crop fires, for example), all of which can affect the amount and timing of agricultural fire. Climate change has and will continue to affect the timing, length, and quality of growing seasons (Porter et al., 2014); the associated impacts on planting and harvest date will affect the timing of crop residue burning, and people will shift the timing of burns to match the shifting phenology of pasture vegetation. It is thus important to understand what information people consider in their decisions of whether, when, and how much to burn. Literature reviews and new research could shed light on indigenous methods for climate forecasting based on changes in the weather and vegetation (e.g., Kagunyu et al., 2016), as well as how these cues might be tied to the timing of prescribed fire for various purposes (e.g., Laris, 2002). Advanced analytical methods could also be applied to climate and fire history observations to look for lagged, region-specific relationships of agricultural burning with weather at weekly to monthly time scales.

While temporal variation is neglected, this first version of FINAL does begin to account for regional variation in agricultural fire management practices. Other aspects of FINAL and LM3, as with many global fire and vegetation models, could be improved by representing such geographic variation. Livestock grazing intensity, as discussed above, is one important example. The shape of the population density-fire relationship also likely varies across the world. Some fire models include a spatially-dependent human ignitions term (Thonicke et al., 2010) to account for this effect. Incorporating this geographic variation into FINAL could improve performance, but it would be important to do so based on independent analyses so as to avoid simply compensating for the model's errors.

**Data availability**

Model code and outputs, along with code for the optimization routine, will be made available by the corresponding author upon request.

**Appendix A: Levenberg-Marquardt algorithm: Implementation**

Our implementation of the Levenberg-Marquardt algorithm (Figure A1) began with a Bash script that set up the files and directories necessary to run the fire model at the 241 points. These points would then be run for 1991–2009 in parallel. Once this first iteration was complete, a Python script calculated the sum of squared errors ($S$) over each gridcell ($c$), year ($y$), and



month ($m$):

$$S = \sum_{c=1}^{241} \sum_{y=2001}^{2009} \sum_{m=1}^{12} (E_{c,y,m} - O_{c,y,m})^2. \tag{A1}$$

Here, $E$ refers to the model-estimated burned area, and $O$ refers to an observation-based estimate of burned area. Specifically, we focused on non-agricultural lands, using as our "observations" estimates generated for each month and year by the method detailed in Rabin et al. (2015) but with $\widehat{F_k}$ estimates restricted to non-negative values. The Python script then generated a new parameter set guess based on the initial values of the parameters and saved a flag telling the Bash script to run the model again with the new guess.

After this and subsequent model runs, another Python script would calculate the associated value of the sum of squared errors ($S_t$) and compare it to the sum of squared errors from the most recently accepted guess ($S^*$). If $S_t < S^*$, the current parameter set guess ($\boldsymbol{\beta_t}$) would be "accepted" and become the new value of $\boldsymbol{\beta^*}$, and $\lambda$ would be decreased. Otherwise, $\boldsymbol{\beta_t}$ would be "rejected," with $\boldsymbol{\beta^*}$ retaining its previous value, and $\lambda$ being decreased. In either case, a new guess would then be generated based on $\boldsymbol{\beta^*}$ and the new value of $\lambda$, the model would be run again, and the process would repeat (Figure A1).

The Python script we developed was based on a MATLAB routine for Levenberg-Marquardt solutions of nonlinear least squares problems called marquardt.m (Nielsen, 2001), further documented in Nielsen (1999). Besides porting it to Python, we made a number of changes to the original code. Some restructuring was related to the fact that the new parameter sets could not be evaluated within Python. Others were to incorporate new features, such as the limited multiplicative damping based on work by Transtrum and Sethna (2012) described above.

Nielsen (2001) uses a somewhat complex method to update $\boldsymbol{\delta}$ after every each iteration (Figure A2). If $S_t \geq S^*$, $\lambda$ is multiplied by a value $\nu$, whose initial value is 2 and is doubled after every rejected guess. If a guess is accepted ($S_t < S^*$), $\nu$ is reset to 2, and $\lambda$ is decreased. We made some changes to the original code as a result of the aforementioned restructuring, with $\lambda$ being reduced as:

$$\lambda = \lambda \times max \left( \frac{1}{3}, 1 - \left[ \frac{S}{dL_{t-1}} - 1 \right]^3 \right) \tag{A2}$$

where

$$dL_{t-1} = \boldsymbol{\delta_{t-1}} \times \left( \lambda \times \boldsymbol{\delta_{t-1}} - \mathbf{J^T} \times \boldsymbol{f}. \right) \tag{A3}$$

Note that there have been many methods proposed over the years for updating the damping parameter in the Levenberg-Marquardt algorithm. These impact the size of the steps the algorithm takes while searching through parameter space, with implications for efficiency. However, the math by which the algorithm determines which direction on each dimension to move is unaffected.

The algorithm has several possible stop conditions. We set a maximum of 300 iterations, which was never reached. The algorithm would also stop if the Python script detected that the gradient was decreasing very slowly:

$$||\mathbf{J^T} \times \boldsymbol{f}||_2 \leq 10^{-15}, \tag{A4}$$

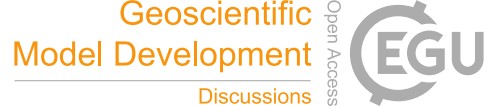



if the step size was very small:

$$||\boldsymbol{\delta_t}||_2 \leq 10^{-15} \times ||\boldsymbol{\beta^*}||_2, \tag{A5}$$

or there was an issue of near-singularity in one of matrices involved in solving for the new parameter step:

$$||\boldsymbol{\delta_t}||_2 \geq \frac{||\boldsymbol{\beta^*}||_2}{\epsilon}, \tag{A6}$$

where $\epsilon$ is the smallest number allowed by the numerical precision of the Python environment. However, in practice, we usually ended up halting the algorithm manually. Each iteration took about two hours, and once we noticed neither the sum of squared errors nor any parameter changing by very much, we would stop the runs. This could have been avoided by choosing more appropriate threshold values for the stop conditions, but likely did not appreciably impact the results.

We initially selected 250 land cells at random from the LM3 grid, but rejected 9 for various reasons (all glacier, all lake, etc.). This left us with 241 gridcells which we would use for the optimization. Preliminary tests, however, revealed a few problems

with the selection: A bias towards improving model fit in gridcells with strong model underestimation was evident (i.e, gridcells where the model simulated too much fire were undersampled), and the high northern latitudes – which make up a small fraction of global land area and an extremely small fraction of global fire activity – were judged to be oversampled. We got rid of 14 of those far northern gridcells (from Greenland and the Canadian tundra), then selected 23 new cells to bring us up to 250. The new cells were specifically selected from cells where a preliminary model run either underestimated or overestimated

non-agricultural burned area relative to the unpacked data. Unfortunately, the model's performance in that preliminary run did not well match how the model actually performed in our optimization run. As such, we ended up oversampling areas of underestimation, leading to a bias towards making the model burn too much. We then culled the most extreme underestimated gridcells one by one until the sums of squared errors from underestimated and overestimated gridcells generated by the initial guess were approximately equal. This left us again with 241 gridcells, whose locations and initial sum of squared errors are

shown in Figure 2a. A histogram of the mean annual error in burned area of the initial guess (Fig. 2b) shows that the positive and negative errors in this new dataset are approximately balanced.

*Author contributions.* All authors contributed to the conceptual design of the model and to editing this manuscript. S.R. and S.M. contributed code. S.R. performed model runs, composed most of this manuscript, analyzed model results, and produced figures. B.M. also contributed to figure design.

*Acknowledgements.* S.R. was supported by a National Science Foundation Graduate Research Fellowship, and by the Carbon Mitigation Initiative. B.I.M. was supported by NSF BCS-1436496. Special thanks to Daniel Ward and Stijn Hantson for their helpful comments.



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

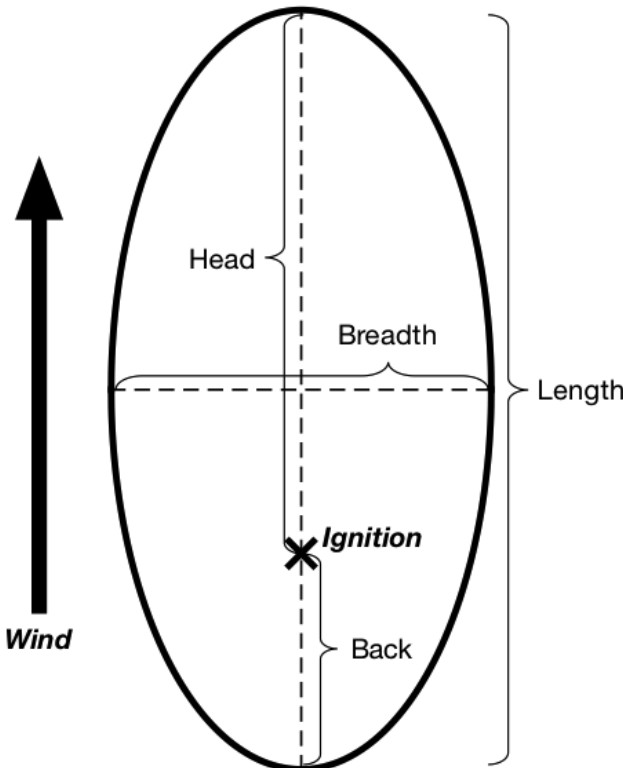

**Figure 1.** Approximation of fire as an ellipse. Adapted from van Wagner (1969) and Arora and Boer (2005).





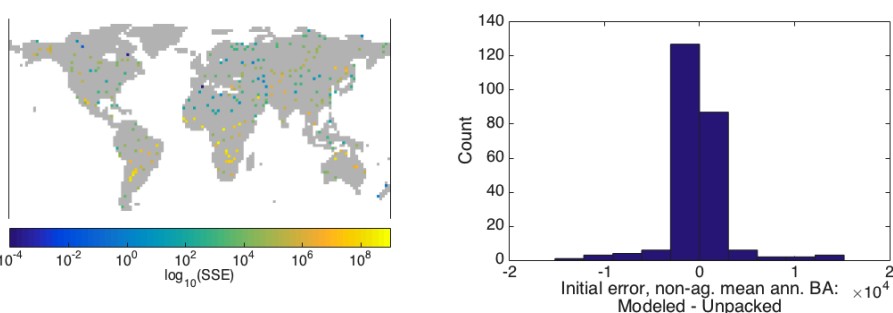

**Figure 2.** Summary of performance of initial guess in gridcells chosen for optimization with regard to non-agricultural burning. **(a)** Map of sum of squared errors. **(b)** Histogram of error in mean annual burned area.

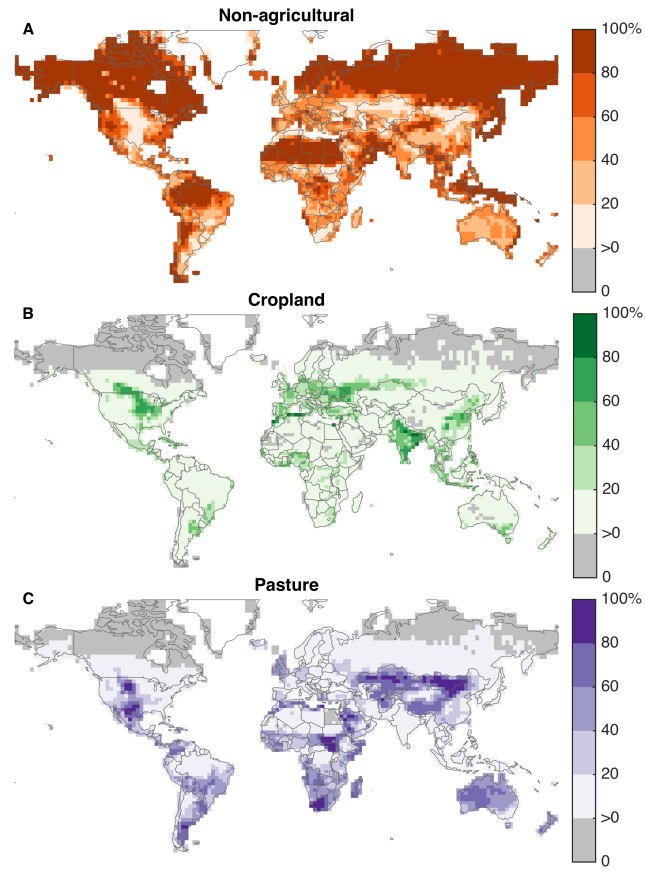

**Figure 3.** Mean fractional land cover of **(a)** non-agricultural land, **(b)**, cropland, and **(c)** pasture over 2001–2009 as simulated in model runs (after Hurtt et al., 2011). Gray cells did not contain any of the indicated land cover type.

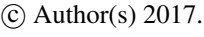



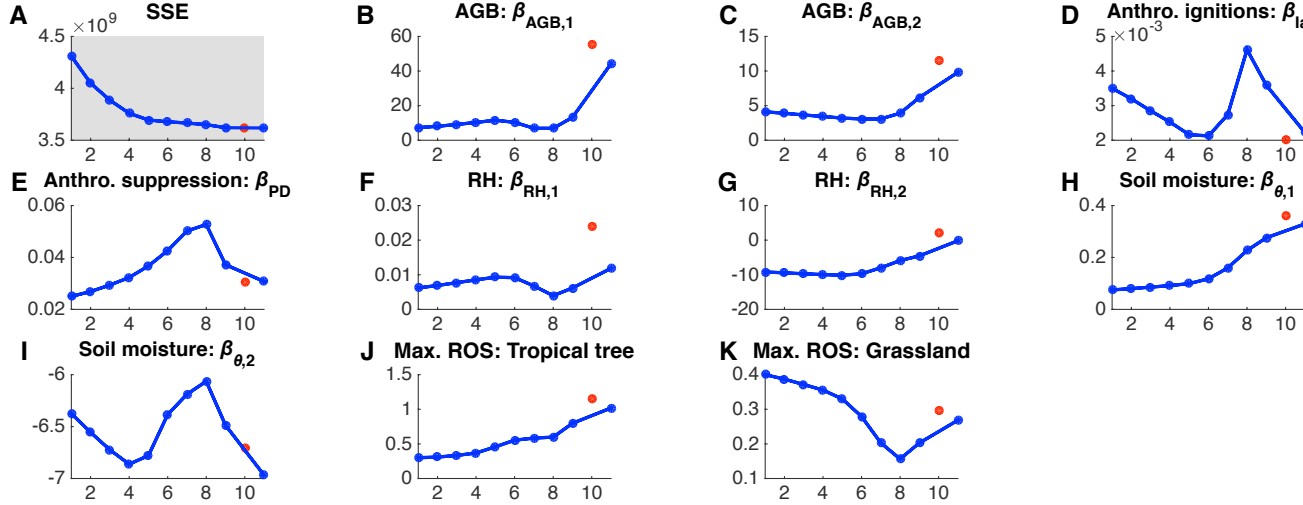

**Figure 4.** Trace plots showing the progression of sum of squared errors **(a)** and each of the ten parameters **(b–k)** over the length of the optimization. X-axes show iteration number, Y-axes show sum of squared errors or parameter guess value, and color of points indicate whether the associated parameter set guess was accepted (blue) or rejected (red).



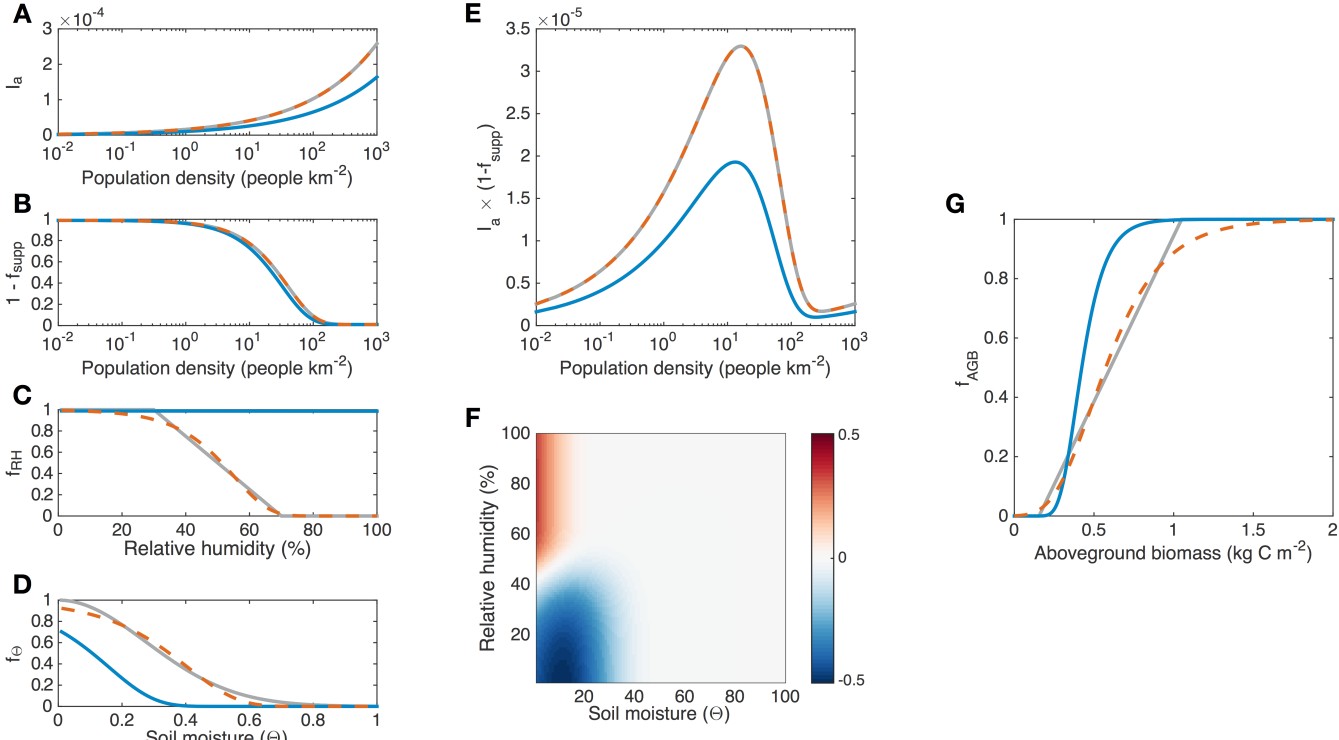

**Figure 5.** Changes in functions that were optimized, from original Li et al. (2012, 2013) functions (solid gray) to initial guesses with Gompertz-style functions where necessary (dashed red) to final parameter set (solid blue). Color bar in panel **f** indicates difference in the cubed product of $f_\theta$ and $f_{RH}$ (range $0 - 1$) between the original and new parameterizations, with blue indicating a lower value in the new parameterization.

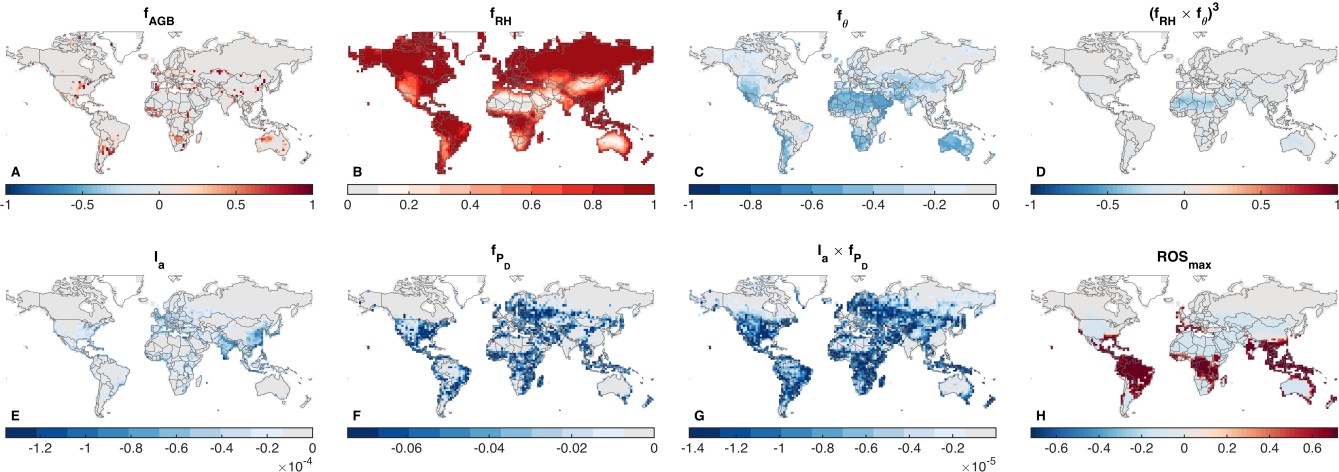

**Figure 6.** Difference in mean value of various fire model functions over 2001–2009 between `FINAL_V0` and `FINAL_V1`. Red indicates regions where the function in `FINAL_V1` allows more fire than in `FINAL_V0`; blue, less.



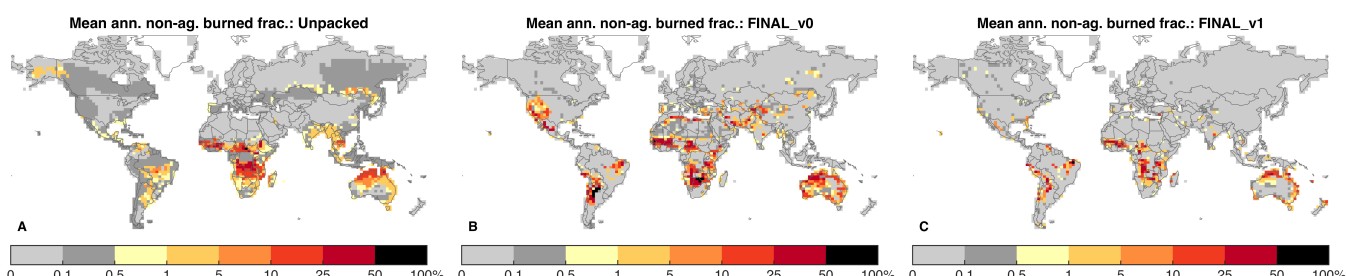

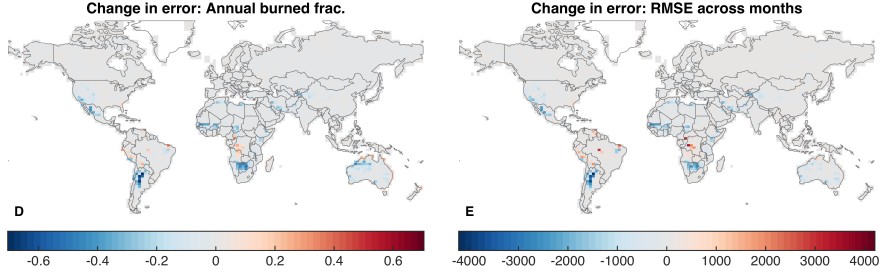

**Figure 7.** Improvement in non-agricultural fire model performance between the initial guess (run `FINAL_V0`) and the final parameter set (run `FINAL_V1`). **(a–c)** Mean annual burned fraction on non-agricultural lands from unpacking **(a)**, the initial guess **(b)**, and the final parameter set **(c**; identical to Fig. 8i.) **(d–e)** Difference between runs `FINAL_V0` and `FINAL_V1` in correspondence of modeled to unpacked non-agricultural burning as measured by mean annual burned fraction **(d)** and root mean squared error evaluated at monthly resolution **(e)**. For **(d)** and **(e)**, blue indicates improvement by `FINAL_V1` over `FINAL_V0`.



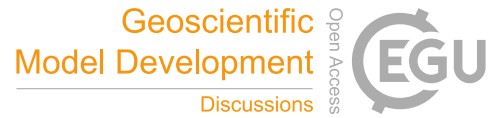

**Figure 8.** Mean annual burned fraction over 2001–2009. (**a**): From GFED3s (Randerson et al., 2012); (**b–e**): observational estimates from unpacking analysis; (**f–i**): Model-estimated. ([i] is identical to Fig. 7c.)





**Figure 9.** Absolute error in mean annual burned fraction **(a–d)** and fire carbon emissions **(e–h)** for each land cover type: Model-estimated minus observational estimates from unpacking analysis.





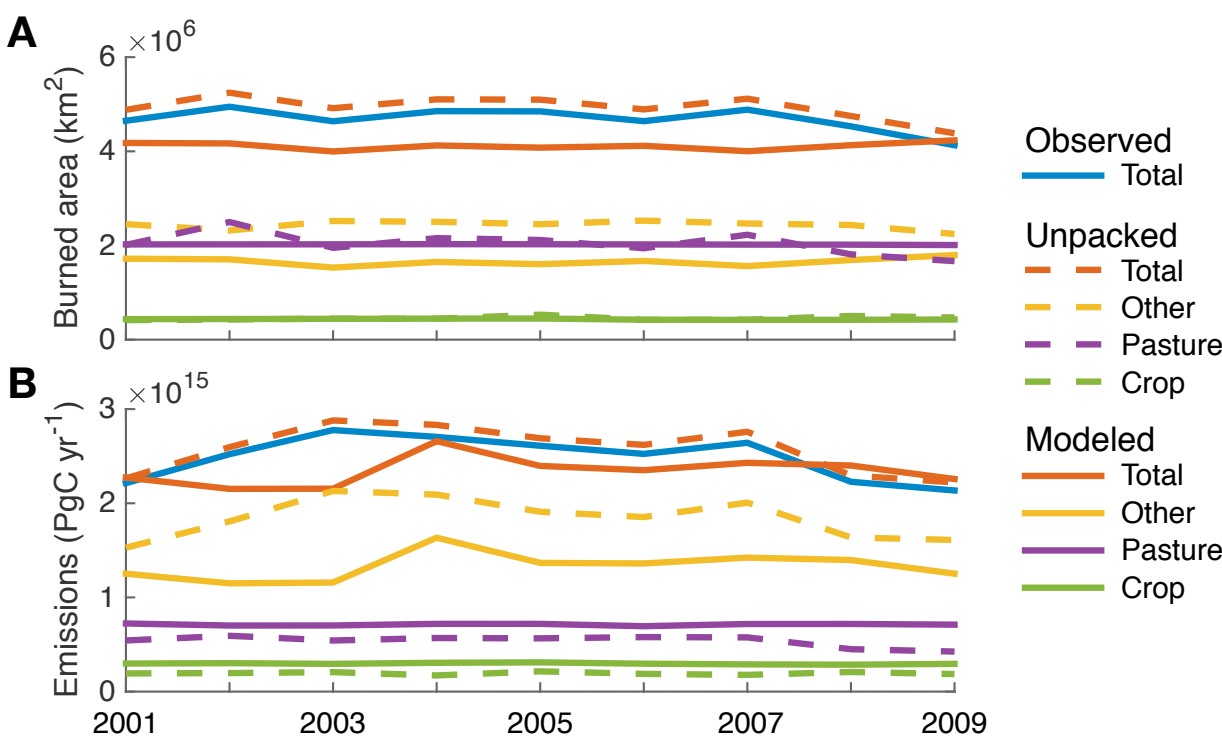

**Figure 10.** Annual time series of observed and model-estimated burned area (**a**, km²) and fire carbon emissions (**b**, PgC yr⁻¹) from 2001–2009. Dashed lines: Observational estimates of total and by-landcover fire emissions from Rabin et al. (2015). Solid blue lines: Observations of total emissions from GFED3s (Randerson et al., 2012). Other solid lines: Model-estimated total and by-landcover fire emissions.





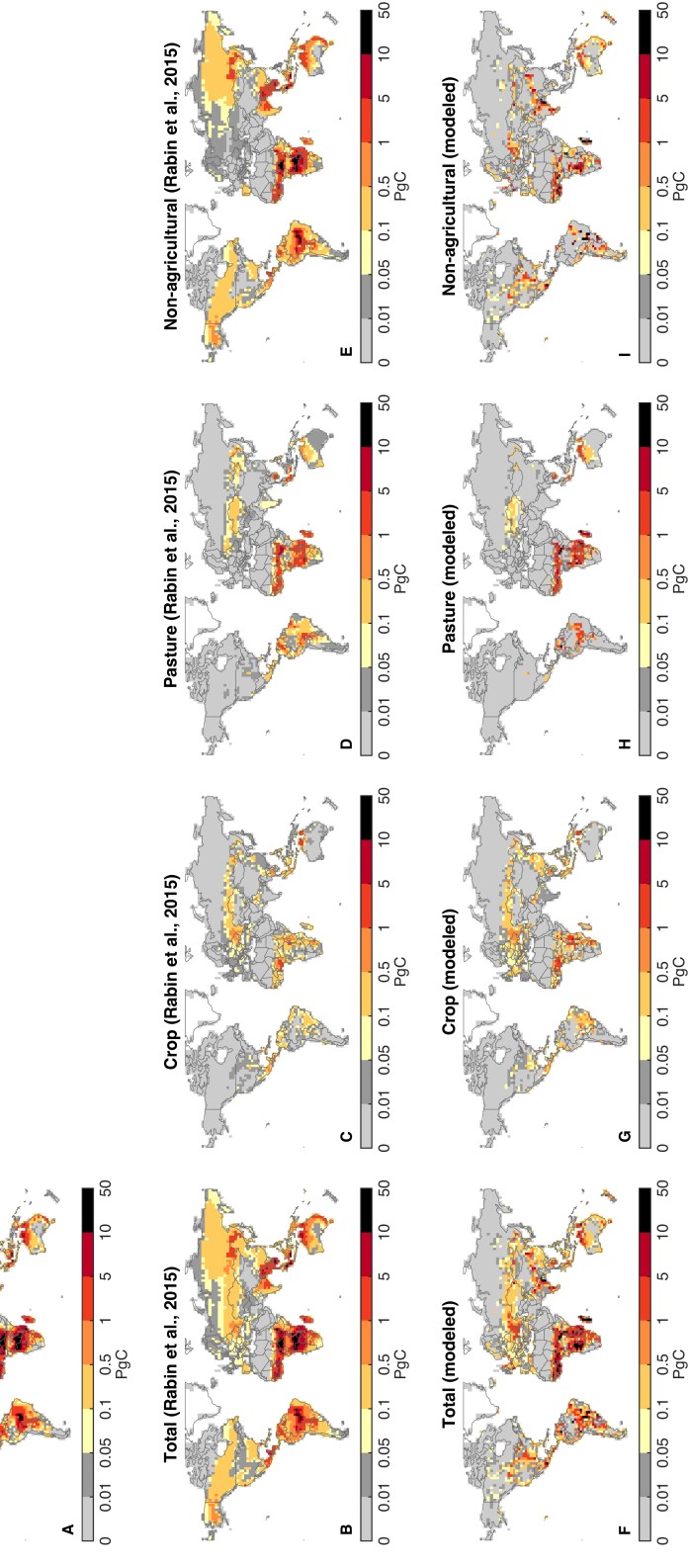

**Figure 11.** Mean annual fire carbon emissions. (**a**): From GFED3s (Randerson et al., 2012); (**b–e**): observational estimates from unpacking analysis; (**f–i**): Model-estimated.)

**Figure 12.** Mean timing of peak burned area (**a**): From GFED3s (Randerson et al., 2012); (B–E): observational estimates from Rabin et al. (2015); (F–I): Model-estimated. Unburned cells in each map are colored gray. Tick marks and labels placed on the 15th of each month.



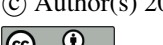

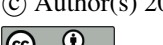

**Figure 13.** Coefficient of variation (standard deviation divided by mean) of non-agricultural burned fraction in $6 \times 6$ gridcell kernels ($12°$ latitude $\times$ $15°$ longitude). **(a)** Modeled; **(b)** from artificially-constructed GFED3s non-agricultural fire data as described in text; **(c)** unpacked. Note log scale of color bars.



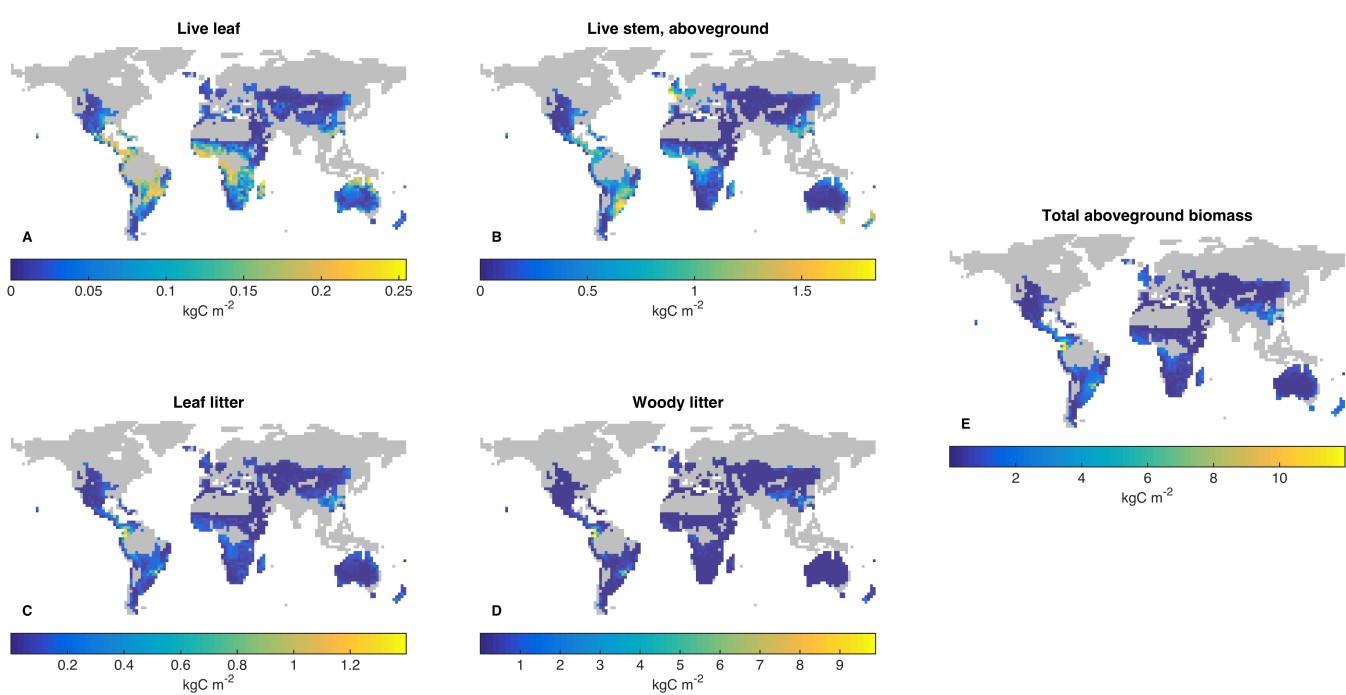

**Figure 14.** Mean aboveground carbon pools on pasture over 2001–2009. Gridcells composed of <20% pasture are shown in gray; note that color scales differ between sub-figures. **(a)**: Live leaves; **(b)**: aboveground live stem; **(c)**: leaf litter; **(d)**: woody litter; **(e)**: total aboveground biomass.

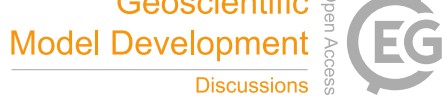



**Table 1.** Combustion completeness and mortality values for each "species" and tissue pool. Note that "stem" refers to both aboveground and belowground stem biomass, and that "root" refers only to fine roots.

| Species | Combustion completeness | | | | Mortality | | | |
| --- | --- | --- | --- | --- | --- | --- | --- | --- |
| | Leaf | Stem | Root | Litter | Leaf | Stem | Root | Litter |
| C4 grass | 0.85 | 1.00 | 0.00 | 0.85 | 0.85 | 0.00 | 0.20 | n/a |
| C3 grass | 0.85 | 1.00 | 0.00 | 0.85 | 0.85 | 0.00 | 0.20 | n/a |
| Tropical tree | 0.70 | 0.15 | 0.00 | 0.50 | 0.70 | 0.60 | 0.10 | n/a |
| Temperate deciduous tree | 0.70 | 0.10 | 0.00 | 0.45 | 0.70 | 0.55 | 0.07 | n/a |
| Evergreen tree | 0.75 | 0.20 | 0.00 | 0.55 | 0.75 | 0.65 | 0.13 | n/a |



**Table 2.** Experimental runs performed in this study. "1–300" indicates that 300 years were simulated, but that these are not tied to any historical data, and thus they do not correspond to any actual historical years. "Graze rate" refers to the amount of non-wasted leaf biomass consumed each day by livestock on pasture tiles.

| Name | Fire model | Years | Initial conditions | Climate | $CO_2$ | Land use | Graze rate | Non-agri. fire: Humans | Agri. fire |
|---|---|---|---|---|---|---|---|---|---|
| LM3_ORIG | Original | "1–300" | Cold start | Repeated 1948–1977 | 286 ppm | Off | n/a | n/a | n/a |
| | | 1861–1947 | — | Repeated 1948–1977 | Historical | Historical | 0.07% | n/a | n/a |
| | | 1948–1991 | — | Historical | Historical | Historical | 0.07% | n/a | n/a |
| FINAL_V0 | New fire model structure; initial guess parameter set | 1948–2009 | As LM3_ORIG | Historical | Historical | Historical | 4% | On | As unpacked |
| FINAL_V1 | New fire model structure; optimized parameter set | 1948–2009 | As LM3_ORIG | Historical | Historical | Historical | 4% | On | As unpacked |





**Table 3.** Values of each optimized parameter, before (Initial) and after (Final) optimization.

|  | Initial | Final |
|---|---|---|
| $\beta_{RH,1}$ | 0.0062 | 0.011856898 |
| $\beta_{RH,2}$ | $-9.1912$ | $-0.172544308$ |
| $\beta_{\theta,1}$ | 0.0750 | 0.329099402 |
| $\beta_{\theta,2}$ | $-6.3741$ | $-6.967427375$ |
| $\beta_{AGB,1}$ | 7.3157 | 44.20896443 |
| $\beta_{AGB,2}$ | 4.11 | 9.820100287 |
| $\beta_{Ia,m}$ | 0.0035 | 0.002224368 |
| $\beta_{PD}$ | 0.025 | 0.030732082 |
| $\beta_{ROSgr}$ | 0.4 | 0.268421539 |
| $\beta_{ROStt}$ | 0.3 | 1.018599996 |

**Table 4.** Global mean annual burned area and associated carbon emissions, 2001–2009. `FINAL_V0` and `FINAL_V1` refer to experimental runs (Table 2). T: Total; C: Cropland; P: Pasture; O: Other land.

|  | Burned area ($10^6$ km$^2$ yr$^{-1}$) | | | | C emissions (PgC yr$^{-1}$) | | | |
|---|---|---|---|---|---|---|---|---|
|  | T | C | P | O | T | C | P | O |
| GFED3s | 4.68 | 0.332[A] | — | — | 2.48 | n.d. | — | — |
| Unpacked | 4.93 | 0.454 | 2.04 | 2.44 | 2.57 | 0.194 | 0.538 | 1.84 |
| `FINAL_V0` | 6.38 | 0.434 | 2.02 | 3.93 | 2.21 | 0.295 | 0.703 | 1.21 |
| `FINAL_V1` | 4.11 | 0.434 | 2.02 | 1.66 | 2.34 | 0.297 | 0.712 | 1.33 |

(A) Midpoint of values for cropland burning with (0.208) and without (0.456) including cropland-natural mosaic.





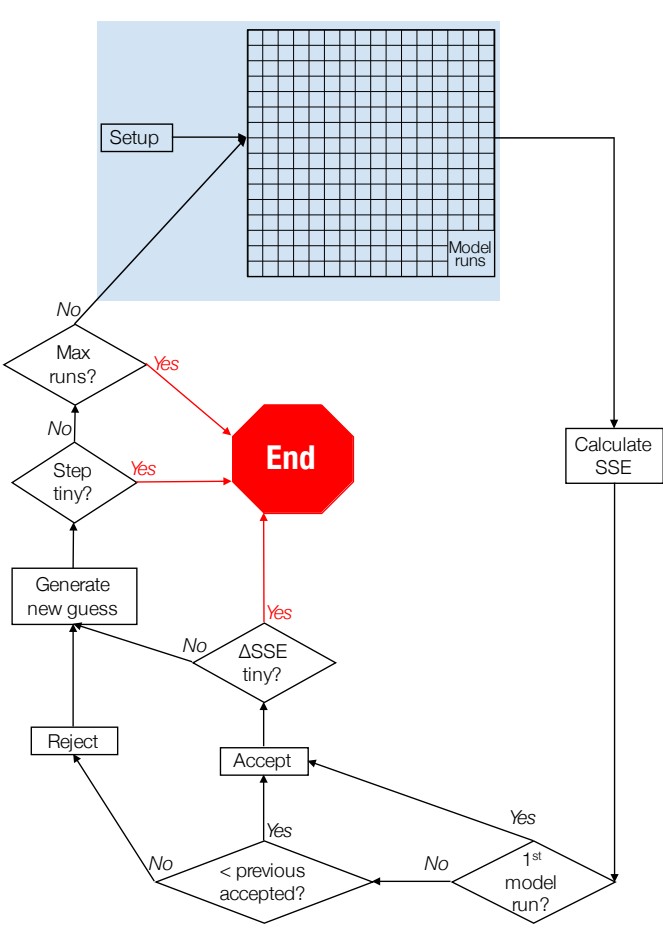

**Figure A1.** Flowchart describing our implementation of the Levenberg-Marquardt algorithm. Blue shading indicates operations related to running the model; all other steps occur in Python.





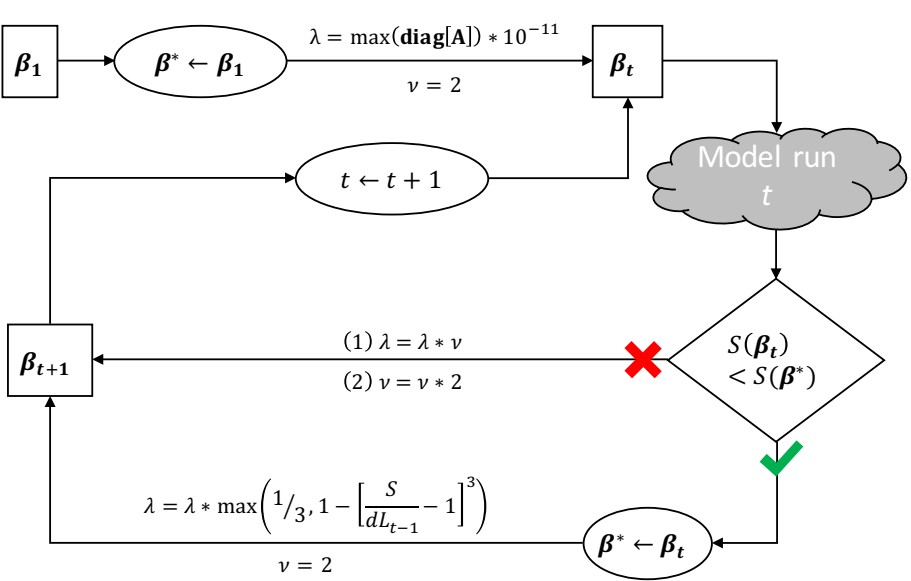

**Figure A2.** Method for updating λ, after Nielsen (1999) and Nielsen (2001).