# Peer review of "A fire model with distinct crop, pasture, and non-agricultural burning: Use of new data and a model-fitting algorithm for FINAL.1"

_Geoscientific Model Development, 2017_

## Author Comment (AC1) · 2 May 2017

There is no technical reason why we could not publish our code in some kind of repository. However, at the time of submission, the code has not yet been "cleaned up" (i.e., commented, trimmed of unused functions, etc.) to the extent that we would be comfortable sharing it without a personal point of contact. By the time of final publication we hope to have the fire model code more generally available.

---

## Author Comment (AC2) · 11 May 2017

The fire model and optimization code are now available on GitHub with the DOI 10.5281/zenodo.574451 (link: http://dx.doi.org/10.5281/zenodo.574451). If the manuscript passes peer review, we will edit the "Data Availability" section accordingly.

Only the fire module part of LM3 is included because of ongoing discussions within GFDL about the best way to distribute the complete codebase. For now, LM3 is accessible via the instructions for AM3 at GFDL's website: https://www.gfdl.noaa.gov/am3/. This URL will also be included in the "Data Availability" section.

---

## Referee Comment (RC1) · Anonymous Referee #1 · 2 Jun 2017

In this paper, the authors described the Fire Including Natural & Agricultural Lands model (FINAL), a fire module for the LM3 land model. One of the most important features of this model is the explicit separation between non-agricultural, pasture and cropland fires : this is a very important feature since fire seasonality is expected to differ significantly between these different fire category.

In the FINAL model, the fraction of cropland and pasture fires is directly estimated from the Rabin et al. 'unpacked' dataset, and the modelling of non-agricultural fires is based on the CLM fire module. This modul is clearly described in the article, along with the modifications done by the authors to adapt it the the LM3 land model. The parameters

of the model, which are expected to be different from those of the CLM module, are determined with an optimization method : this optimization relies on the Levenberg-Marquardt algorithm, which minimizes the sum of squared errors between the model and the GFED3s data, for a selected sample of grid cells. The authors took care to ensure that all functions involved in the models were continuously differentiable, which is mandatory to perform such an optimization.

Because non-natural fires are directly estimated from burned area data, simulated non-natural burned area is very close to the results from Rabin et al. 2015. The results are not as good for non-natural fires, probably resulting from the strong limitation induced by soil moisture after the optimization of parameters. The results of the model, along with its limitations, are well-discussed in the article, and the authors proposed an interesting critical discussion about the optimization process. However, I still have some questions concerning the implementation of the optimization method, which need some clarifications. They are listed in the Specific Comments part.

Specific Comments :

1) You stopped the optimization after 11 steps, and said (lines 21-22, page 14) : 'By the eleventh iteration, it did not seem that allowing iterations to continue would result in much improved sums of squared errors'. I have some major concerns here. First, I think you should put the SSE subplot on Figure 4 in log scale, since the range is driven by the SSE values during the first steps and does not allow to clearly see what's happening after the fourth step. It is very common that during an optimization process, the function to minimize drops very quickly during the first steps, and then need some time to finally converge. Second, looking at the evolution of the other parameters, it is not so clear that the algorithm converged : the parameters vary more when the difference of squared errors Delta_SSE between two steps vary less. I would really like to see 4-5 supplementary steps, to see if the parameters reach a state of stability, and to ensure that the SSE is really stable after this number of step.
2) If I understand it correctly, your optimization is only done on 241 grid cells, as described in Appendix A. I think that the last paragraph of the Appendix should be included as a section 2.6.3., since it is very important for the reader to know this as he reads the methodology section, and not when he reached the discussion part : before reaching it, I thought you did the optimization on all the grid cells. I suppose this allows you to run the model much faster, but you said in the discussion : 'The deeply model-interactive setup used here – where the complete model of soil, vegetation, and fire was forced with climatic data for 19 model years – took around two hours per iteration with all gridcells being run in parallel'. But if you run the model on a limited number of cells, shouldn't it be faster ? If it is not possible to run the model only with a fixed selection of cells, then why don't you compute the SSE on a much higher number of cells ? I think you should give a clearer explanation on this choice in the article.

3) I think an important consistency check would be to specifically look at the squared errors of these selected indivual cells after the minimization process (as a second map on figure 2 for example, and, even better if you can, an histogram of the difference of SSE before/after the minimization). This will also allow to clearly check if the optimization process is mainly driven by savannas/grassland, where a small change of parameters will have huge effect on the modeled burned area, hence on the SSE in this cell (as you said in the discussion part).

4) Section 2.6.2 : not all the parameters of the model are involved in the minimization process. If it seems clear why you have chosen to optimized the parameters Beta_Ia, Beta_ROS and Beta_ROS, it is not the case for the remaining parameters. I think the authors should explicit why the have choosen these parameters (the ones driven by soil moisture), and not, for example, those driven by the temperature.

Technical comments :

1) One of the strength of the FINAL model comes from the separation of agricultural/pasture and natural fires. I think it should be more emphasized in the article. To

do so, I suggest to move the discussion about the 'unpacked' input data in section 3.2 to section 2.3. I also think that it is necessary to explain clearly what is the Fk fraction (equation 1 from Rabin et al. 2015 could appear in the article), since it is necessary to understand how the fire types are separated in Rabin et al. 2015.

2) If you decide to use capital letters to reference the figure, you should also use capital letters when you mention it in the caption or in the text. Moreover, it would be clearer if the letters were close to the titles of the subfigures.

3) Concerning the colorbar on the Figures 7,8 and 11 : I really think you should replace the dark grey (the color corresponding to 0.1 < BA < 0.5 for example) with a color 'yellow-ish' color, I think it hides too much the cells with low but non-negligible burned area fraction.

4) I think you can remove Figure 1. It is not really usefull, and there are already lots of figures.

5) Figure 12 : There is no map background for the month map, it should be added for the sake of homogeneity with other figures.

6) In figure 5 (which, I think, is really nice) : I didn't find the definition of f_supp, but I supposed that fPD = 1 - fsupp. If this is the case, I think you should either put f_PD as the axe label in Figure 5b, or explicitly write the relation between fPD and fsupp somewhere, for the sake of clarity.

7) In Table 3 : the final values should have the same number of digits as the initial values. You could even put the difference (or percentage of variation) between the two sets of value as a third column.

8) In general, there are lots of map in the article. I understand it is necessary to show separate maps for non-agricultural/pasture/cropland, but maybe you could, for example, remove the total map from Rabin et al. 2015, or the one from GFED3s, in all the figures. I do not have a strong opinion on this last item, I just think that it is easier

for the reader to focus on a smaller number of plots.

Please also note the supplement to this comment:
http://www.geosci-model-dev-discuss.net/gmd-2017-77/gmd-2017-77-RC1-supplement.pdf

––––––––––––––––––––––––––––––––

---

## Referee Comment (RC2) · Anonymous Referee #2 · 7 Jun 2017

This paper presents a new fire model FINAL (Fire Including Natural & Agricultural Lands model) which simulates fires on managed agricultural land as distinct from non-agricultural fires. These managed fires are further separated into types of land-use: cropland and pasture management fires. This is an important development for fire modelling because, as the authors correctly point out, there are very few fire models that currently distinguish between agricultural and non-agricultural fires, and even fewer that separate cropland fires from pasture fires. One of the main reasons for this has been a lack of observational data, but the recent development of estimated burned area datasets for cropland, pasture and non-agricultural land by Rabin et al (2015) has now made it possible to incorporate this information into fire modelling. The dynamic

global vegetation model LM3 is used, with the fire model for non-agricultural land based on Li et al (2012, 2013), and the agricultural fire model based on gridded climatology maps from Rabin et al (2015) unpacking analysis of monthly estimates of burned area.

It is my opinion that this paper presents a relevant advance in modelling science within the scope of GMD, which leads the way for future studies reviewing the contribution of agricultural fires to total burned area and emissions. The paper presents a novel way of using new data from Rabin et al (2015) to model fires within a DGVM. The methods of modelling non-agricultural fires after Li et al (2012, 2013) are clearly outlined along with the relevant equations, and it is stated where they have moved away from Li et al methods to, for example, Gompertz functions and why. Later in the paper there is a detailed explanation of the parameter optimization used for the non-agricultural fires to adapt it to LM3. There is also a clear description of the set-up for the experimental runs. Now the code has also been made available on GitHub, the description of methods seems comprehensive and reproducible.

There is a fairly thorough presentation of results and analysis of the model, including improvements from FINAL v0 to v1, the mean burned area and carbon emissions compared to GFED data and the unpacking analysis data, presented spatially and temporally. These support the evaluation and conclusions made in the paper. There is one appendix including two figures, describing the implementation of the Levenberg-Marquardt algorithm, which seems appropriate in content and length.

The title accurately reflects the content of the paper, and the abstract gives a good summary of what the model does, what is new about the approach, and highlights the key results of the model in simulating the amount, distribution, and timing of burnt area and emissions. Agricultural fire simulations are very close to the unpacked data from Rabin et al (2015), which is to be expected because the data were used to force the model over crop and pasture areas, but the results for non-agricultural fires are less closely matched to observations. The authors present an excellent discussion on why this might be the case, and make suggestions for future work to improve the

model. Overall the paper is presented well, with fluent language and a clear and logical structure.

Specific comments:

It would be nice to see a fuller discussion about how large the contribution of land-use / agricultural fires is in the introduction, to give some context to how important this is and why it is necessary to breakdown fires into crop, pasture and non-agricultural categories. As a regional example, Xie et al ('Dynamic Monitoring of Agricultural Fires in China from 2010 to 2014 using MODIS and GlobeLand30 Data, 2016) showed that agricultural burning in China accounts for 60% of all fire activity in the last 5 years.

The observational data used was from GFED3s. Whilst it is an improvement that GFED3s was chosen over GFED3 to include small fires, can the authors explain why the latest dataset GFED4s which also includes the contribution from small fires was not used?

I am not an expert in the optimization method used, so will leave others to comment on this.

Technical comments:

Double check equation 7; from the Li et al (2012) paper $- \pi$ is used, although this is not used for any calculations here so is purely a typographical comment

On page 16 line 14, you state 'Pasture fire did not experience such severe error in burned fraction anywhere (Fig. 9d)', after pinpointing the two errors in figure 9c over one European gridcell and over several gridcells in Northern Australia. At first it seemed as though you were overlooking the errors in pasture burning in Europe, SE Asia and across Australia. Then I spotted the 'x10-3' in between the plots, which must correspond to the bottom plot, although this is quite hidden. Perhaps it is worth also pointing out these error points, but also making clear that the scale for the pasture plot is different.

Page 16 line 20; I think the reference here should be to figures 8e and 8i, not 8b and 8f

Page 23 refers to FINALv1 being represented in an ESM, but in the introduction it states that the offline DGVM version of LM3 was used. I assume there would be further work needed to couple this into the ESM, so this statement is not quite accurate

I believe figure 2 is not referenced in the paper until the Appendix. Considering there are already a lot of figures, perhaps this should be moved and added to the list of figures in the Appendix

In some of the figures the term 'Non-agricultural fires' (figures 7, 8, 9, 11) is used, and in some 'Other' is used (figures 10, 12). It would be better if this was made consistent across the figures

Figures 9 uses a different order of presenting results (total, non-agriculture, crop, pasture), to 10 (total, non-agriculture, pasture crop,) and to 11 & 12 (total, crop, pasture, non-agriculture). As with (5), it would be better if this was made consistent across the figures.

---

## Referee Comment (RC3) · Anonymous Referee #3 · 13 Jun 2017

**General comments**

The authors describe a novel fire module FINAL to the DGVM LM3 which distinguishes fires on cropland, on pasture and on non-agricultural areas in terms of driving conditions and development. Thus, the approach allows to separate fire-related emissions in terms of seasonal occurrence and cause. The procedure is well documented with the modifications applied to previous work by Li et al., coherently laid out, well structured and very understandable, especially the discussion.

Nevertheless, I have one major concern which is centered around the parameter optimization. Here, three obstacles hinder rapid publication which are partly acknowledged

by the authors:

- The chosen algorithm may find a local minimum instead of the global one. The criterion for convergence is not clearly defined.

- For the global model only a selection of grid points is chosen for the calibration procedure without information on the selection criteria. In case of undersampling climatic conditions, the resulting parameter set may not be ideal for the neglected region or the influence of one of the drivers may be underestimated because this variable did vary between the chosen grid cells.

- The error metric is already discussed in the manuscript. It would be good to at least complement the metric by others especially designed for comparison of model results and observations.

The mentioned flaws in the design of the optimization lead to a parameter set that extinguishes one of the drivers for fire occurrence namely relative humidity. The authors should motivate the chosen method in a way that this result is convincing and the reader is not suspecting it to be caused by making an inappropriate choice. The neglect of relative humidity while strengthening the role of soil moisture usually asks for the correlation of these drivers. Please make clear why and in which way both variables play a role.

**Specific comments**

P1L13: 'the boreal zone suffers from underestimates', please rephrase because it is unlikely that the boreal zone really suffers.
P3L25: the argument that an MCMC approach would be too costly is understandable but maybe worthwhile when the parameter space really has to be explored. There are also other approaches like the version using generations which could help.
P4L12: 'state-of-the art' -> 'state-of-the-art'

P5L16 and Eq. 4 and nearly all further equations: inconsistency of brackets. There should be round brackets for functions and square brackets for indices. You use both for the same expression which is disturbing.

P11L6: 'all N sample gridcells selected for the optimization'. How many grid cells were selected, how and why? What are the criteria for this?

P12L4: why were parameters from eq 12, 13 or 20 not selected for optimization?

P13L2: the symbol $F_k$ is not explained before

P13L12: the resolution of LM3 could be mentioned earlier in the general description.

P14L21: The optimization process takes only 10 time steps. The criteria for convergence remain completely unclear and the parameter value development makes it unclear if there was a convergence. This part of the approach should be included in methods and the convergence decision should be motivated.

Fig. 5: shows clearly that fire suppression by relative humidity is gone completely but that by soil moisture is even stronger. Also population density gets a stronger influence and that of AGB becomes less with the resulting parameter set. This is mentioned in the discussion but in the results it does not become clear why this parameter set should be accepted.

P15L11: the question on an substitutional effect of soil moisture and relative humidity arises again. Could you comment on that?

P17L30: the figure may be moved to the appendix and only the numbers be included in the text to describe differences in the spatial heterogeneity.

P18L8: long-lasting fires are an interesting topic and only mentioned briefly. Could you include a short comment on expected improvements or if you intend to investigate this further?

P19L6 to L28: this part could be moved to the results.

P20L32: this is critical because you undermine your resulting parameter set. How is the artifact possible? Could it be caused by the choice of the grid cells for optimization? Why should the reader accept the chosen parameters?

P22L3: this is an interesting information. Which input data are additionally used and

why are they not further taken into account? Do they also result in a suppression of the factor relative humidity?

P22L15: after this reasoning it is even more important to consider at least more time steps in the optimization procedure or to consider a different parameter space search.

P22L17: this is a good discussion on the error metric. Please consider to complement SSE by other metrics also in the result section (e.g. see R package QualV; https://www.jstatsoft.org/article/view/v022i08)

P23L18: this valuation is refreshing in its clarity and honesty but please consider the effect on the reader. Are you really not convinced that the chosen approach was successful? In this case, the optimization has to be redone with a different selection procedure for the reduced gridcell set, a different optimization algorithm and an increased number of simulations.
* * *

---

## Author Comment (AC3) · 22 Nov 2017

We thank the editors and reviewers for their help in improving this manuscript.

The reviewers' suggestion that the optimization appeared not to have finished was especially helpful. We re-ran the optimization (four times, with slightly randomized initial conditions) and allowed it to continue much longer than before. The burned area and emissions performance of the final parameter set did change much, but some features (such as the relative humidity and soil moisture functions) turned out to be more reasonable than before.

Note that the new optimization and model runs would not have been possible without the help of Daniel S. Ward, who we would like to add as second author.

The length of the main text has been reduced by moving some figures to a new Supplementary PDF, as well as by moving the discussion about pasture biomass to the Appendix (B).

As mentioned in a previous comment, information about accessing the fire model and optimization code has now been provided in the Data Availability section.

Attached, find a ZIP file with new versions of the main text and the new supplementary PDF. Included as well is a PDF containing our responses to reviewers and the output of the latexdiff command between the original and new versions of the main text.

Please also note the supplement to this comment:
https://www.geosci-model-dev-discuss.net/gmd-2017-77/gmd-2017-77-AC3-supplement.zip

---

## Author Response (AR1)

**Authors' response**

**Reply to Reviewer Comment 1**

In this paper, the authors described the Fire Including Natural & Agricultural Lands model (FINAL), a fire module for the LM3 land model. One of the most important features of this model is the explicit separation between non-agricultural, pasture and cropland fires : this is a very important feature since fire seasonality is expected to differ significantly between these different fire category.

In the FINAL model, the fraction of cropland and pasture fires is directly estimated from the Rabin et al. 'unpacked' dataset, and the modelling of non-agricultural fires is based on the CLM fire module. This modul is clearly described in the article, along with the modifications done by the authors to adapt it the the LM3 land model. The parameters of the model, which are expected to be different from those of the CLM module, are determined with an optimization method : this optimization relies on the LevenbergMarquardt algorithm, which minimizes the sum of squared errors between the model and the GFED3s data, for a selected sample of grid cells. The authors took care to ensure that all functions involved in the models were continuously differentiable, which is mandatory to perform such an optimization.

Because non-natural fires are directly estimated from burned area data, simulated nonnatural burned area is very close to the results from Rabin et al. 2015. The results are not as good for non-natural fires, probably resulting from the strong limitation induced by soil moisture after the optimization of parameters. The results of the model, along with its limitations, are well-discussed in the article, and the authors proposed an interesting critical discussion about the optimization process. However, I still have some questions concerning the implementation of the optimization method, which need some clarifications. They are listed in the Specific Comments part.

**Specific Comments :**

1) You stopped the optimization after 11 steps, and said (lines 21-22, page 14) : 'By the eleventh iteration, it did not seem that allowing iterations to continue would result in much improved sums of squared errors'. I have some major concerns here.

First, I think you should put the SSE subplot on Figure 4 in log scale, since the range is driven by the SSE values during the first steps and does not allow to clearly see what's happening after the fourth step. It is very common that during an optimization process, the function to minimize drops very quickly during the first steps, and then need some time to finally converge.

Plotting SSE on a log scale here makes little difference; the values along the Y-axis vary within a single order of magnitude. We have, however, added a subplot to the former Fig. 4 (now Fig. 2) showing the relative improvement in SSE between accepted parameter sets, plotted on a log scale. The following is now in the text: "After an initial drop in SSE over the first six guesses, subsequent guesses did not result in much improvement, with SSE not differing by more than 0.001% between accepted guesses after the 19th iteration (Fig. 2a-b)."

Second, looking at the evolution of the other parameters, it is not so clear that the algorithm converged : the parameters vary more when the difference of squared errors Delta\_SSE between two steps vary less. I would really like to see 4-5 supplementary steps, to see if the parameters reach a state of stability, and to ensure that the SSE is really stable after this number of step.

Thanks to this comment and a similar one from Reviewer 3, we let this optimization continue. It is now referred to as Optimization 1 in the text. We also added three more optimization runs. Optimization 1 ended up not being stable at the parameter values chosen in the initial manuscript—it actually wasn't stable at all, instead veering off into model-breaking parameter space for one of the relative humidity parameters. We have settled on the result of Optimization 3 as the "canonical" parameter set; for more details, see Section 4.1 (optimization results).

2) If I understand it correctly, your optimization is only done on 241 grid cells, as described in Appendix A. I think that the last paragraph of the Appendix should be included as a section 2.6.3., since it is very important for the reader to know this as he reads the methodology section, and not when he reached the discussion part : before reaching it, I thought you did the optimization on all the grid cells. I suppose this allows you to run the model much faster, but you said in the discussion : *'The deeply model-interactive setup used here – where the complete model of soil, vegetation, and fire was forced with climatic data for 19 model years – took around two hours per iteration with all gridcells being run in parallel'.* But if you run the model on a limited number of cells, shouldn't it be faster ? If it is not possible to run the model only with a fixed selection of cells, then why don't you compute the SSE on a much higher number of cells ? I think you should give a clearer explanation on this choice in the article.

We have added the following paragraph to Section 2.6.1:

Briefly, we ran the model for 1991--2009 in a sample of 241 gridcells. A Python script evaluated the model performance and suggested a new parameter set, which was fed back into the model. The Python script then checked the performance of the new parameter set, accepted that set if its performance was improved relative to the previous set, and generated a new guess. This process continued until the routine encountered at least five rejected parameter set guesses. We did not optimize over all gridcells because of computational limitations; even with all 241 gridcells being run in parallel, each iteration of the optimization took around two hours. More details on our implementation of the algorithm, including how the gridcells in the sample were selected, can be found in Appendix A.

What we meant in the text quoted by the Reviewer above was that the *optimization* takes approximately two hours per iteration with all 241 gridcells being run in parallel. We have amended "all gridcells" in the quoted section to read "all 241 gridcells". We apologize for the lack of clarity.

3) I think an important consistency check would be to specifically look at the squared errors of these selected indivual cells after the minimization process (as a second map on figure 2 for example, and, even better if you can, an histogram of the difference of SSE before/after the minimization). This will also allow to clearly check if the optimization process is mainly driven

by savannas/grassland, where a small change of parameters will have huge effect on the modeled burned area, hence on the SSE in this cell (as you said in the discussion part).

We have added optimization-gridcell-specific maps and bar graphs for Optimizations 3 and 2 as Figures S2 and S3, respectively. Grasslands/shrublands do appear to have exerted the most influence on the optimization, but only in the subtropics and temperate zone. Tropical savannas generally experienced worsened performance after optimization. This is now noted in the last paragraph of Section 4.1.

4) Section 2.6.2 : not all the parameters of the model are involved in the minimization process. If it seems clear why you have chosen to optimized the parameters Beta\_Ia, Beta\_ROS and Beta\_ROS, it is not the case for the remaining parameters. I think the authors should explicit why the have choosen these parameters (the ones driven by soil moisture), and not, for example, those driven by the temperature.

Temperature does not limit flammability most of the time in most gridcells. Indeed, the gridcells with the largest influence on the parameterization—tropical savanna regions—are never affected by  $f_T$ . This means that the  $T_{lo}$  and  $T_{up}$  parameters (former Eq. 11, now 13) would not be well-constrained in the analysis; thus, we did not include them.

We did not include the parameters affecting the upper and lower asymptotes of  $f_{PD}$  (former Eq. 12, now 14) because we were already optimizing two parameters governing the effect of population density on number of fires ( $\beta_{Ia,m}$  and  $\beta_{PD}$ ). We decided to limit the degrees of freedom with regard to the combined population density functions.

We did not optimize parameters in the former Eq. 13 (governing the effect of wind speed on fire length:breadth ratio; now Eq. 15) or Eq. 20 (governing the effect of decreasing burnable area on maximum fire size; now Eq. 22) in the interest of limiting somewhat the scope of our optimizations. The parameters in these equations are generally based on phenomena external to the CLM model used by Li et al. (2012, 2013)—Eq. 13 (15) is derived from equations used by the Canadian Forest Service (Arora & Boer, 2005), and Eq. 20 (22) is derived from hypothetical simulations performed by Pfeiffer et al. (2013) independent of any fire or vegetation model.

We have added a paragraph explaining these decisions to the end of Section 2.6.2.

Technical comments :

1) One of the strength of the FINAL model comes from the separation of agricultural/pasture and natural fires. I think it should be more emphasized in the article. To do so, I suggest to move the discussion about the 'unpacked' input data in section 3.2 to section 2.3. I also think that it is necessary to explain clearly what is the Fk fraction (equation 1 from Rabin et al. 2015 could appear in the article), since it is necessary to understand how the fire types are separated in Rabin et al. 2015.

**These changes have been made.**

2) If you decide to use capital letters to reference the figure, you should also use capital letters when you mention it in the caption or in the text. Moreover, it would be clearer if the letters were close to the titles of the subfigures.

This has been corrected.

3) Concerning the colorbar on the Figures 7,8 and 11 : I really think you should replace the dark grey (the color corresponding to 0.1 < BA < 0.5 for example) with a color 'yellow-ish' color, I think it hides too much the cells with low but non-negligible burned area fraction.

**The light gray at the low end of these color scales has been lightened, increasing the contrast with the dark gray. These are now Figs. 4, 5, and 8, respectively.**

**4) I think you can remove Figure 1. It is not really usefull, and there are already lots of figures. Fig. 1 has been moved to the Supplement and is now Fig. S1.**

5) Figure 12 : There is no map background for the month map, it should be added for the sake of homogeneity with other figures.

We were unable to find a way to plot the map overlay on this figure in a way that (a) allowed the map lines to be visible across the large swaths of dark color, and (b) did not obscure the mapped data. Note: This figure has been moved to the Supplement and is now Fig. S5.

6) In figure 5 (which, I think, is really nice) : I didn't find the definition of f\_supp, but I supposed that fPD = 1 - fsupp. If this is the case, I think you should either put f\_PD as the axe label in Figure 5b, or explicitly write the relation between fPD and fsupp somewhere, for the sake of clarity.

**fsupp is now explicitly defined as part of the former Eq. 12 (now Eq. 14).**

7) In Table 3 : the final values should have the same number of digits as the initial values. You could even put the difference (or percentage of variation) between the two sets of value as a third column.

**Table 3 now has the same level of precision used for all elements. Table S1 has been added to show the full precision of each element.**

8) In general, there are lots of map in the article. I understand it is necessary to show separate maps for non-agricultural/pasture/cropland, but maybe you could, for example, remove the total map from Rabin et al. 2015, or the one from GFED3s, in all the figures. I do not have a strong opinion on this last item, I just think that it is easier for the reader to focus on a smaller number of plots.

We have moved the former Figures 1, 6, 12, and 13 to the Supplement; these are now Figs. S1, S4, S5, and S6, respectively. We have also moved Figures 2 and 14 to the Appendix; they are now Figs. A3 and A4, respectively.

**Reply to Reviewer Comment 2**

This paper presents a new fire model FINAL (Fire Including Natural & Agricultural Lands model) which simulates fires on managed agricultural land as distinct from nonagricultural fires. These managed fires are further separated into types of land-use: cropland and pasture management fires. This is an important development for fire modelling because, as the authors correctly point out, there are very few fire models that currently distinguish between agricultural and non-agricultural fires, and even fewer that separate cropland fires from pasture fires. One of the main reasons for this has been a lack of observational data, but the recent development of estimated burned area datasets for cropland, pasture and non-agricultural land by Rabin et al (2015) has now made it possible to incorporate this information into fire modelling. The dynamic global vegetation model LM3 is used, with the fire model for non-agricultural land based on Li et al (2012, 2013), and the agricultural fire model based on gridded climatology maps from Rabin et al (2015) unpacking analysis of monthly estimates of burned area.

It is my opinion that this paper presents a relevant advance in modelling science within the scope of GMD, which leads the way for future studies reviewing the contribution of agricultural fires to total burned area and emissions. The paper presents a novel way of using new data from Rabin et al (2015) to model fires within a DGVM. The methods of modelling non-agricultural fires after Li et al (2012, 2013) are clearly outlined along with the relevant equations, and it is stated where they have moved away from Li et al methods to, for example, Gompertz functions and why. Later in the paper there is a detailed explanation of the parameter optimization used for the non-agricultural fires to adapt it to LM3. There is also a clear description of the set-up for the experimental runs. Now the code has also been made available on GitHub, the description of methods seems comprehensive and reproducible.

There is a fairly thorough presentation of results and analysis of the model, including improvements from FINAL v0 to v1, the mean burned area and carbon emissions compared to GFED data and the unpacking analysis data, presented spatially and temporally. These support the evaluation and conclusions made in the paper. There is one appendix including two figures, describing the implementation of the LevenbergMarquardt algorithm, which seems appropriate in content and length.

The title accurately reflects the content of the paper, and the abstract gives a good summary of what the model does, what is new about the approach, and highlights the key results of the model in simulating the amount, distribution, and timing of burnt area and emissions. Agricultural fire simulations are very close to the unpacked data from Rabin et al (2015), which is to be expected because the data were used to force the model over crop and pasture areas, but the results for non-agricultural fires are less closely matched to observations. The authors present an excellent discussion on why this might be the case, and make suggestions for future work to improve the model. Overall the paper is presented well, with fluent language and a clear and logical structure.

**Specific comments:**

It would be nice to see a fuller discussion about how large the contribution of landuse / agricultural fires is in the introduction, to give some context to how important this is and why it

is necessary to breakdown fires into crop, pasture and non-agricultural categories. As a regional example, Xie et al ('Dynamic Monitoring of Agricultural Fires in China from 2010 to 2014 using MODIS and GlobeLand30 Data, 2016) showed that agricultural burning in China accounts for 60% of all fire activity in the last 5 years.

The observational data used was from GFED3s. Whilst it is an improvement that GFED3s was chosen over GFED3 to include small fires, can the authors explain why the latest dataset GFED4s which also includes the contribution from small fires was not used?

I am not an expert in the optimization method used, so will leave others to comment on this.

Technical comments:

Double check equation 7; from the Li et al (2012) paper –  $\pi$  is used, although this is not used for any calculations here so is purely a typographical comment

**This has been corrected (now Eq. 9).**

On page 16 line 14, you state 'Pasture fire did not experience such severe error in burned fraction anywhere (Fig. 9d)', after pinpointing the two errors in figure 9c over one European gridcell and over several gridcells in Northern Australia. At first it seemed as though you were overlooking the errors in pasture burning in Europe, SE Asia and across Australia. Then I spotted the 'x10-3' in between the plots, which must correspond to the bottom plot, although this is quite hidden. Perhaps it is worth also pointing out these error points, but also making clear that the scale for the pasture plot is different.

This has been fixed by using, e.g., 0.005 instead of  $5 \times 10^{-3}$ . The figure is now Fig. 6.

Page 16 line 20; I think the reference here should be to figures 8e and 8i, not 8b and 8f This has been corrected; the figure is now Fig. 5.

Page 23 refers to FINALv1 being represented in an ESM, but in the introduction it states that the offline DGVM version of LM3 was used. I assume there would be further work needed to couple this into the ESM, so this statement is not quite accurate

**This has been corrected.**

I believe figure 2 is not referenced in the paper until the Appendix. Considering there are already a lot of figures, perhaps this should be moved and added to the list of figures in the Appendix We have moved Figure 2 to the Appendix; it is now Figure A3.

In some of the figures the term 'Non-agricultural fires' (figures 7, 8, 9, 11) is used, and in some 'Other' is used (figures 10, 12). It would be better if this was made consistent across the figures

This has been corrected by changing "Other" to "Non-agricultural" in the former Figs. 10 and 12. Note that the figure numbers have changed:  $7 \rightarrow S8$ ,  $8 \rightarrow 5$ ,  $9 \rightarrow 6$ ,  $10 \rightarrow 7$ ,  $11 \rightarrow 8$ ,  $12 \rightarrow S5$ .

Figures 9 uses a different order of presenting results (total, non-agriculture, crop, pasture), to 10 (total, non-agriculture, pasture crop,) and to 11 & 12 (total, crop, pasture, non-agriculture). As with (5), it would be better if this was made consistent across the figures.

The former Figs. 9 and 10 have been made consistent with the former Figs. 11 and 12. Note that the figure numbers have changed:  $9 \rightarrow 6$ ,  $10 \rightarrow 7$ ,  $11 \rightarrow 8$ ,  $12 \rightarrow S5$ .

**Reply to Reviewer Comment 3**

**General comments**

The authors describe a novel fire module FINAL to the DGVM LM3 which distinguishes fires on cropland, on pasture and on non-agricultural areas in terms of driving conditions and development. Thus, the approach allows to separate fire-related emissions in terms of seasonal occurrence and cause. The procedure is well documented with the modifications applied to previous work by Li et al., coherently laid out, well structured and very understandable, especially the discussion.

Nevertheless, I have one major concern which is centered around the parameter optimization. Here, three obstacles hinder rapid publication which are partly acknowledged by the authors:

- The chosen algorithm may find a local minimum instead of the global one. The criterion for convergence is not clearly defined.
- For the global model only a selection of grid points is chosen for the calibration procedure without information on the selection criteria. In case of undersampling climatic conditions, the resulting parameter set may not be ideal for the neglected region or the influence of one of the drivers may be underestimated because this variable did vary between the chosen grid cells.
- The error metric is already discussed in the manuscript. It would be good to at least complement the metric by others especially designed for comparison of model results and observations

The mentioned flaws in the design of the optimization lead to a parameter set that extinguishes one of the drivers for fire occurrence namely relative humidity. The authors should motivate the chosen method in a way that this result is convincing and the reader is not suspecting it to be caused by making an inappropriate choice. The neglect of relative humidity while strengthening the role of soil moisture usually asks for the correlation of these drivers. Please make clear why and in which way both variables play a role.

**Specific comments**

P1L13: 'the boreal zone suffers from underestimates', please rephrase because it is unlikely that the boreal zone really suffers.

**We have changed "suffers from" to "sees".**

P3L25: the argument that an MCMC approach would be too costly is understandable but maybe worthwhile when the parameter space really has to be explored. There are also other approaches like the version using generations which could help.

A costlier computational method might indeed be worthwhile, but the sheer scale of that—thousands of iterations at two hours per iteration—makes it infeasible at this time. We also believe that the manuscript presents an innovative application of Levenberg-Marquardt; although it may not be ideal, the presentation of its benefits and downsides in this manuscript should be valuable to Earth system modelers with a variety of interests.

**P4L12: 'state-of-the art' -> 'state-of-the-art' **This has been corrected.**

P5L16 and Eq. 4 and nearly all further equations: inconsistency of brackets. There should be round brackets for functions and square brackets for indices. You use both for the same expression which is disturbing.

The suggested style is standard syntax for, e.g., Python, but it is not required by the Copernicus style guide (URL below). We prefer to alternate round and square brackets in equations, for easier tracking of where a given bracketed section begins and ends. https://www.geoscientific-modeldevelopment.net/for\_authors/manuscript\_preparation.html

P11L6: 'all N sample gridcells selected for the optimization'. How many grid cells were selected, how and why? What are the criteria for this?

This is explained in Appendix A. A note directing the reader to that Section for details on the sampling procedure has been added to Section 2.6.1.

P12L4: why were parameters from eq 12, 13 or 20 not selected for optimization?

We did not include the parameters affecting the upper and lower asymptotes of  $f_{PD}$  (former Eq. 12, now 14) because we were already optimizing two parameters governing the effect of population density on number of fires ( $\beta_{Ia,m}$  and  $\beta_{PD}$ ). We decided to limit the degrees of freedom with regard to the combined population density functions.

We did not optimize parameters in the former Eq. 13 (governing the effect of wind speed on fire length:breadth ratio; now Eq. 15) or Eq. 20 (governing the effect of decreasing burnable area on maximum fire size; now Eq. 22) in the interest of limiting somewhat the scope of our optimizations. The parameters in these equations are generally based on phenomena external to the CLM model used by Li et al. (2012, 2013)—Eq. 13 (15) is derived from equations used by the Canadian Forest Service (Arora & Boer, 2005), and Eq. 20 (22) is derived from hypothetical simulations performed by Pfeiffer et al. (2013) independent of any fire or vegetation model.

We have added a paragraph explaining these decisions to the end of Section 2.6.2.

P13L2: the symbol Fk is not explained before

A more thorough discussion of the method used in Rabin et al. (2015) is now included in Section 2.3.

P13L12: the resolution of LM3 could be mentioned earlier in the general description. **It is now mentioned in Section 2.1.**

P14L21: The optimization process takes only 10 time steps. The criteria for convergence remain completely unclear and the parameter value development makes it unclear if there was a convergence. This part of the approach should be included in methods and the convergence decision should be motivated.

Thanks to this comment and a similar one from Reviewer 1, we let this optimization continue. It is now referred to as Optimization 1 in the text. We also added three more optimization runs. Optimization 1 ended up not being stable at the parameter values

**chosen in the initial manuscript—it actually wasn't stable at all, instead veering off into model-breaking parameter space for one of the RH parameters. We have settled on the result of Optimization 3 as the "canonical" parameter set; for more details, see Section 4.1 (optimization results).**

Fig. 5: shows clearly that fire suppression by relative humidity is gone completely but that by soil moisture is even stronger. Also population density gets a stronger influence and that of AGB becomes less with the resulting parameter set. This is mentioned in the discussion but in the results it does not become clear why this parameter set should be accepted.

With the results from Optimization 3, we no longer see the extreme result vis a vis the relative humidity and soil moisture functions. Of course, there are still changes in most functions, which we do not discuss in detail in the Results section. We believe covering these in the Discussion section (specifically Sect. 5.3) makes more sense, as it allows a clearer separation between the objective results of the optimization (in Results) and our interpretation of them (in Discussion). We have added a note near the beginning of Section 4.1 to appropriately set the reader's expectations about the division of material between the Results and Discussion sections.

P15L11: the question on an substitutional effect of soil moisture and relative humidity arises again. Could you comment on that?

With the results from Optimization 3, this is no longer an issue.

P17L30: the figure may be moved to the appendix and only the numbers be included in the text to describe differences in the spatial heterogeneity.

The former Fig. 13 has been moved to the Supplement and is now Fig. S6.

P18L8: long-lasting fires are an interesting topic and only mentioned briefly. Could you include a short comment on expected improvements or if you intend to investigate this further?

The following sentence has been added to the end of that paragraph: "A new version of FINAL, FINAL.2, does include multi-day fire, and is successfully able to reproduce the distribution of fire frequency binned by duration in boreal Canada. However, even with that and other changes impacting fire behavior in the boreal zone, FINAL.2 still does underestimate burned area there (Ward et al., in review)."

P19L6 to L28: this part could be moved to the results.

We believe this passage is better suited to the Discussion, as we delve into the simulation of pasture biomass only in an effort to understand why pasture emissions are so high. Discussing pasture biomass in the Results would make more sense in the context of an evaluation of LM3's performance with regard to biomass generally—a discussion outside the scope of this manuscript.

P20L32: this is critical because you undermine your resulting parameter set. How is the artifact possible? Could it be caused by the choice of the grid cells for optimization? Why should the reader accept the chosen parameters?

With the results from Optimization 3, this is no longer an issue.

P22L3: this is an interesting information. Which input data are additionally used and why are they not further taken into account? Do they also result in a suppression of the factor relative humidity?

This text has been clarified; it now reads, "We also used a different source for climate forcing data and calibrated our model based on different burned area data." With the result from Optimization 3,

P22L15: after this reasoning it is even more important to consider at least more time steps in the optimization procedure or to consider a different parameter space search.

As discussed above, we have now extended our initial optimization run and added three more.

P22L17: this is a good discussion on the error metric. Please consider to complement SSE by other metrics also in the result section (e.g. see R package QualV; https://www.jstatsoft.org/article/view/v022i08)

**The other metrics presented in that article are very interesting and potentially useful, but unfortunately not realistic to be included in this manuscript due to the time it would take to fully understand and learn them.**

P23L18: this valuation is refreshing in its clarity and honesty but please consider the effect on the reader. Are you really not convinced that the chosen approach was successful? In this case, the optimization has to be redone with a different selection procedure for the reduced gridcell set, a different optimization algorithm and an increased number of simulations.

We have changed that paragraph to read:

The choice of gridcells and initial conditions is also extremely important to any automated model fitting algorithm. We strove to maximize model robustness by experimenting with different initial parameter set guesses (Knorr et al., 2014; Le Page et al., 2015). A more structured and informed approach to sampling gridcells for the optimization – and increasing the number of gridcells – would further improve robustness.

**A fire model with distinct crop, pasture, and non-agricultural burning: Use of new data and a model-fitting algorithm for FINALv1FINAL.1**

Sam S. Rabin1,2, Daniel S. Ward3, Sergey L. Malyshev3, Brian I. Magi4, Elena Shevliakova3, and Stephen W. Pacala1

[revised manuscript text omitted]